# Smart-Plexer: a breakthrough workflow for hybrid development of multiplex PCR assays

Luca Miglietta [1,2], Yuwen Chen[2,3], Zhi Luo [1,3], Ke Xu[1,2], Ning Ding[1], Tianyi Peng[2], Ahmad Moniri[2], Louis Kreitmann[1], Miguel Cacho-Soblechero[2], Alison Holmes[1], Pantelis Georgiou[2] & Jesus Rodriguez-Manzano [1✉]

Developing multiplex PCR assays requires extensive experimental testing, the number of which exponentially increases by the number of multiplexed targets. Dedicated efforts must be devoted to the design of optimal multiplex assays ensuring specific and sensitive identification of multiple analytes in a single well reaction. Inspired by data-driven approaches, we reinvent the process of developing and designing multiplex assays using a hybrid, simple workflow, named Smart-Plexer, which couples empirical testing of singleplex assays and computer simulation to develop optimised multiplex combinations. The Smart-Plexer analyses kinetic inter-target distances between amplification curves to generate optimal multiplex PCR primer sets for accurate multi-pathogen identification. In this study, the Smart-Plexer method is applied and evaluated for seven respiratory infection target detection using an optimised multiplexed PCR assay. Single-channel multiplex assays, together with the recently published data-driven methodology, Amplification Curve Analysis (ACA), were demonstrated to be capable of classifying the presence of desired targets in a single test for seven common respiratory infection pathogens.

[1] Department of Infectious Disease, Faculty of Medicine, Imperial College London, London, UK. [2] Department of Electrical and Electronic Engineering, Faculty of Engineering, Imperial College London, London, UK. [3] These authors contributed equally: Yuwen Chen, Zhi Luo. ✉email: j.rodriguez-manzano@imperial.ac.uk

Quantitative Polymerase Chain Reaction (qPCR) enables real-time monitoring of interactions between target-specific oligonucleotides, such as primers and probes, and their target during amplification[1–3]. The extraordinary ease and reliability of this golden standard method for nucleic acid amplification tests (NAATs) have improved routine diagnostics in several fields and, more recently, played a crucial role during the COVID-19 pandemic, one of the 10 deadliest infectious diseases in history[4–6]. This epidemic has highlighted the need for rapid, accurate, cost-effective, simple to use, and ideally Point-of-Care (PoC) multiplex assays to support timely infection management[7–13].

Current screening strategies for multiple pathogens are reported to be expensive, sample-consuming, and in some cases, inaccurate[14–16]. As a result, multiplex PCR is emerging as an inexpensive alternative for multi-target identification[17–20]. Many efforts have been made in developing novel methods to increase the number of targets detected by multiplex assays and to enhance the accurate identification of multiple infectious sources in a single test[21,22]. Advances in multi-pathogen detection include the use of high-resolution melting analysis (HRMA), fluorescent probe-based method, or restriction enzyme digestion[23–25]. Recently, the emergence of machine learning approaches in clinical diagnostics has highlighted the potential of data-driven multiplexing, which, compared to conventional methods, unbars limitations in terms of throughput, costs, time and reliability[26–28]. Methods h using either melting curve analysis (intercalating dye-based chemistries) or final fluorescence intensity (probe-based assays) have been proposed as features for machine learning algorithms[29,30]. Moreover, using cutting-edge signal processing and tailored amplification chemistries, state-of-the-art identification performance has been achieved by leveraging the kinetic information encoded in the entire amplification curve from multiplex PCR assays. A novel learning-based methodology called amplification curve analysis (ACA) has been recently reported as a digital tool to expand multiplex capabilities of real-time PCR-based diagnostic platforms, increasing the number of detectable targets per fluorescent channel in a single reaction without hardware modification[31–34].

However, the development of multiplex PCR assays is restricted by the need for extensive experimental testing to evaluate analytical performance, including cross-reactivity, specificity, and sensitivity[21,35–37]. One of the biggest challenges in multiplexing is the complexity of assay design, which dramatically increases with the number of targets, making the development costly, lengthy and resource-consuming in the wet laboratory[14,38]. This biological problem can be mathematically described as following: for $N_t$ multiplexed targets, if $N_{Ps}$ candidate primer sets are designed for each of them (which is trivial progress for well-designed singleplex assays), the total number of possible multiplex assay combinations is $N_c = N_{Ps}{}^{N_t}$ (e.g. $N_c = 16,384$ when $N_{Ps} = 4$ and $N_t = 7$). The $N_c$ increases exponentially with $N_t$ making it impractical to find the optimal combination for high-level multiplexing by wet-lab experiments. Therefore, in-silico simulation methods could offer fast screening and optimised multiplex design.

To address this problem, here we present the Smart-Plexer, a mathematical algorithm capable of simulating thousands of possible multiplex assay combinations based on singleplex real-time digital PCR (qdPCR) data. We aim to demonstrate the use of this new methodology by developing a TaqMan-based multiplex assay, in a single fluorescent channel, for the specific and sensitive detection of seven common respiratory tract infection (RTI) pathogens. This work is two-folded: First, we validated the Smart-Plexer by comparing the performance of all possible simulated and empirical combinations in 3-plex, showing a strong correlation between in-silico and lab-tested multiplexes; Second, we assessed the proposed pipeline in high-level multiplex (7-plex) by evaluating the ACA classification performance on synthetic DNA and clinical samples. We demonstrated that, out of 4608 simulated combinations, an optimal multiplex assay could be developed using this novel framework to detect seven common respiratory pathogens accurately in qdPCR.

## Results

**Smart-Plexer design framework**. We developed the Smart-Plexer, a framework that uses singleplex PCR reactions as a 'card deck' to generate a 'winning combination' multiplex assay. After defining the number of targets and uploading datasets generated from real-time PCR reactions with a single primer set (or singleplex assay) and a single target, the Smart-Plexer will combine amplification curve data from each target, simulating multiplex assays (Fig. 1). These simulations, representing different singleplex assay combinations, are further validated through wet-lab multiplex tests conducted for each target. This wet-lab multiplex evaluation enables us to assess the changes in the curve shape of the amplification reaction during the transition from a singleplex to a multiplex environment. Therefore, empirical multiplex tests involve running the actual multiplex assay in the laboratory using the selected simulated multiplex primer sets. This allows us to directly observe and analyse the real-time amplification curves and evaluate the performance and accuracy of the multiplex assay (empirical multiplex), providing empirical validation of our approach when multiple primer sets are present.

To identify multiple targets with empirical multiplexes, the Smart-Plexer framework was coupled and evaluated with the ACA methodology. It is a methodology that utilises machine learning techniques to analyses and classify the amplification curves generated in PCR reactions. By capturing the kinetic information encoded in the amplification curves, ACA can effectively differentiate and identify different targets or analytes present in the reaction. It involves extracting relevant features from the amplification curves and employing machine learning algorithms to train a classifier that can accurately classify and distinguish between various targets based on their unique curve characteristics. The ACA recognises clusters from different amplification shapes which in our case represent different targets. Therefore, by using this approach it is crucial to maintain differences among sigmoidal trends in-silico. The difference between the sigmoidal curve of each target is analysed by the Smart-Plexer through distance measurements (such as Euclidian distance). This novel framework is capable of distance calculation from either the entire amplification curve or its sigmoidal features. The average of computed distances among all the targets is used to rank each combination of singleplex (or simulated multiplex) from high to low inter-curve similarity values. Moreover, the ranking system takes the minimum distance between the two closest targets to ensure that the simulated multiplex with high average values is not dependent on the high difference of only a group of curves. When two amplification curves have high similarity, hence a small distance value, the ACA classifier will not work efficiently to identify either target. Therefore, the rank of the combination depends on both average and minimum distance scores. A set of singleplex assays from the top ranks were selected as simulated multiplex for the empirical validation in the laboratory, and the ACA performance was assessed.

To compute distances between amplification curves, the Smart-Plexer requires a filtering process where the amplification data generated undergo the following steps: (i) subtraction of curve background to remove the fluorescence signal noise at the starting

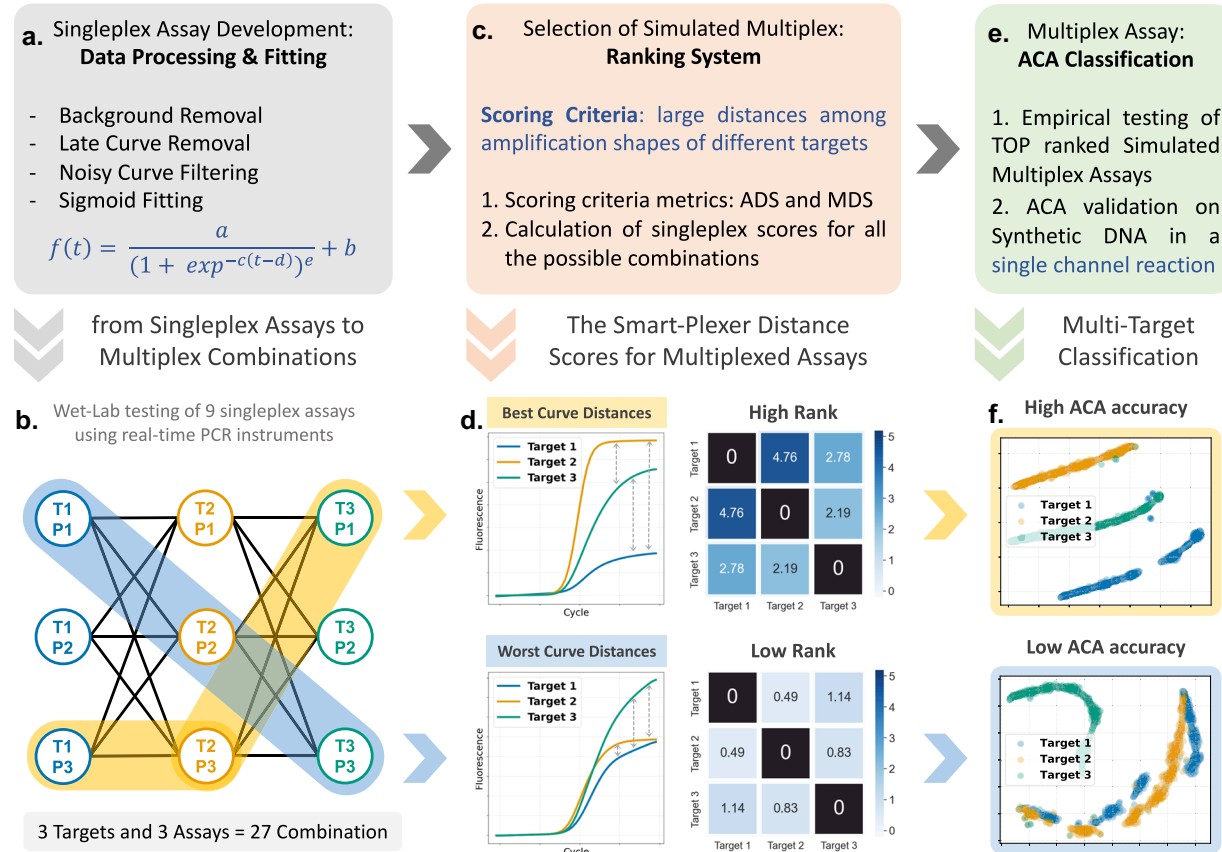

**Fig. 1 Smart-Plexer workflow. a** Given a dataset of singleplex real-time PCR reactions (real-time amplification curves), a processing step is applied. (**a.i–a.iii**) The processed curves are fitted following the equation depicted in step. **a.iv** An example is given in **b**, where each curve resulting from singleplex reactions is used in a simulation of multiplex assays. Three targets are considered, and each of them has three unique singleplex assays (a total of 27 simulated combinations). **c** The simulated multiplex scores are calculated from the Smart-Plexer according to the Scoring Criteria. **d** Distances within curves from different targets are calculated based on mathematical algorithms (such as Euclidean), and as shown in the confusion matrices, resulting values are used to rank multiplex assays from high (high distances within targets) to low (low distances within targets). **e** High-rank multiplex assays are chosen for empirical testing, and the ACA method is used to evaluate the classification performance on target identification of each selected multiplex. **f** Cluster visualisation with 2-D t-SNE represents the difference in inter-target distances between a High-Rank and a Low-Rank multiplex, resulting in high and low ACA classification accuracy, respectively.

cycles, (ii) removal of late amplification curves to exclude non-plateau reactions, (iii) removal of noisy curves to exclude non-sigmoidal shapes resulted by operator or instrumentation faults[39]. The following step comprised a fitting equation using the five-parameter model proposed by Spiess et al.[40].

**Selection of representative amplification curve.** The ACA method uses the entire amplification curve as a time series where fluorescence values change as the number of cycles increases. Firstly, we chose the entire raw amplification curve generated from the real-time PCR reaction as the input of the Smart-Plexer. Secondly, the framework was evaluated using curves normalised with the final fluorescence intensity (FFI) as input to assess performance changes by removing the absolute fluorescence information. To further investigate changes related to different curve representations and different levels of data abstractions (feature dimensions) provided to the Smart-Plexer, sigmoidal parameters generated from a fitting model were also used as input to assess the influence on this framework.

To evaluate the best-fitting model, primary efforts have been focused on the selection of an appropriate equation. Several methods have been proposed to efficiently model the real-time PCR sigmoid, such as four, five, and six-parametric functions[40–42]. As a case study, we retrieved the amplification

curve data previously reported by Moniri et al. (2020)[31]. Using raw curves as input, after sigmoidal fitting, we calculated the mean square error (MSE) between the raw and the fitted curves for the entire dataset. As shown in Supplementary Table 1, the lowest MSE is achieved with the five-parametric model (MSE = 0.0036). The rising MSE in six-parameter sigmoid fitting is caused by unsuccessful optimisation resulting from a larger searching dimension. Based on the lowest MSE value, it is determined to utilise the five-parameter sigmoid function to extract features, and the equation is given below:

$$f(t) = \frac{a}{\left(1 + \exp^{-c(t-d)}\right)^e} + b \qquad (1)$$

where $t$ is the amplification time (or PCR cycle), $f(t)$ is the fluorescence at time $t$, $a$ is the maximum fluorescence, $b$ is the baseline of the sigmoid, $c$ is related to the slope of the curve, $d$ is the fractional cycle of the inflection point, and $e$ allows for an asymmetric shape (Richard's coefficient).

The three different curve representations (raw curves, FFI normalised curves and fitted parameters) were further used to evaluate the transferability from singleplex to multiplex reactions in the Smart-Plexer.

**Average distance score (ADS) and minimum distance scores (MDS) based on curve distances to rank multiplex assays**. We developed two distance metrics to measure transferability from simulated to empirical multiplexes, since it is hypothesised that distances between amplification curves should be maintained during the transition from singleplex to multiplex.

It is possible to calculate distances between two distinct curves by considering them as two data points in the multidimensional space and quantify their distances using various metrics (i.e., Euclidian, Cosine, and Manhattan). In a single-channel multiplex assay, the number of primer sets present in the reaction equals the number of targets ($N_t$), therefore the number of distances ($N_d$) among curves of different targets is represented by the following formula:

$$N_d = \binom{N_t}{2} = \frac{N_t(N_t-1)}{2} \qquad (2)$$

The average of all the distances is used to assign a score to the multiplex assay called average distance score (ADS). The ADS provides information on the overall distances across targets, and the higher its values are, the more distant the curves are, and better ACA performance is expected (as distances are related to data point clusters). A high ADS does not guarantee a large distance between every two targets of the multiplex. To overcome this limitation, we considered a second metric called minimum distance score (MDS), the distance value of the two closest curves (minimum value of the given $N_d$ distances).

The ADS and MDS narrow down the selection of empirical testing for the highest-performing multiplexes using a ranking system. Moreover, they are used to validate that inter-curve distance information is maintained during the transition from simulated to empirical multiplexes, and they can be used to develop assays in silico more suitable for ACA, skipping costly and timely laboratory testing.

**Smart-Plexer validation using a 3-plex assay**. To assess the performance of the Smart-Plexer for both in-silico multiplex development and ACA classification accuracy, we designed three primer sets for three selected targets: Adenovirus (HAdV), Human coronavirus HKU1 (HCoV-HKU1) and the Middle East respiratory syndrome-related coronavirus (MERS-CoV). The primers were evaluated using synthetic DNA and tested by real-time digital PCR (qdPCR). As shown in Fig. 1, the number of combinations to test using $N_t$ targets ($N_t = 3$) and $N_{Ps}$ assays for each target ($N_{Ps} = 3$) is 27 ($N_c = N_{Ps}^{N_t} = 27$ combinations, listed in Supplementary Table 2). Three targets were chosen to validate the Smart-Plexer because a complete comparison of all the 27 simulated and empirical multiplex assays can be experimentally conducted as the number of wet-lab experiments is achievable ($N_c \times N_t = 81$ tests).

The wet-lab testing of each primer set (or singleplex assay) was conducted, and the resulting raw data were combined in a total of 27 simulated multiplexes as explained before. Similarly, experiments were carried out on combinations of primer sets (or empirical multiplex assays) in a single-channel reaction. A group of amplification curves, which can be considered as data points in multidimensional spaces, were generated from a unique interaction between each assay and its specific target. Supplementary Fig. 1 illustrates the raw curve considered in this experiment and Supplementary Table 3 shows the curve counts and the Ct variation among them. The median of these data points was calculated to represent each group of curves. Furthermore, distances among all the curve medians were used to generate the ADS and MDS of all the possible combinations Fig. 2a, b visually represent the correlation between the in-silico and wet-

lab tested assays using ADS and MDS in simulated and empirical multiplexes. Pearson coefficients were reported for both ADS as 0.301, 0.972 and 0.607, and MDS as 0.092, 0.761, and 0.686, for raw curve, normalised curve, and fitted parameters, respectively (visual representations of each curve type/parameters are depicted in Fig. 2c, and ADS and MDS for all the curve types/combinations are reported in Supplementary Table 4).

It can be observed that normalised curve correlations scored higher than the rest in both ADS and MDS, showing that simulated and empirical multiplex are correlated if FFI is discarded. It is also important to note that the use of all the five curve parameters worsens the correlation as the bimodal distribution of parameter "$e$" negatively influences the correlation, as discussed by Miglietta et al. (2022)[39]. Moreover, the correlation from singleplex to multiplex might be affected by the fact that the "$d$" parameter is related to the cycle threshold ($C_t$) of the amplification curve. Target concentration can be influenced by instrumentation, operator, and experimental errors; therefore, variabilities of $C_t$ can easily mislead the correlation of the five parameters using "$d$". Moreover, the scope of conducting this correlation is to compare purely sigmoidal shapes, and concentrations of the nucleic acid targets should not affect the distance values of the two curves. In addition, the use of parameters "$a$" and "$b$" is redundant as: (i) "$a$" is related to the FFI, and as shown in the middle plot of Fig. 2a, b, FFI is not relevant to the distance correlation and (ii) all curves present in this dataset were processed with a background removal (baseline correction) and all "$b$" parameters were levelled to almost zero.

The observed correlation between simulated and empirical multiplex distances promoted the identification of representative features which would maintain the information of distances during the translation from a singleplex to a multiplex environment. As mentioned before, the parameter "$a$", "$b$", "$d$" and "$e$" can negatively influence the correlation for both ADS and MDS; therefore, we focus on the "$c$" parameter.

**The key parameter for curve distances correlation in multiplex assays: the "slope"**. The previous section reported all the correlation coefficients for ADS and MDS between simulated and empirical multiplexes for different curve representations: raw curves, normalised curves, and fitting parameters. Both ADS and MDS showed the maximum correlation values when considering normalised curves. Those results, along with our discussion on the fitted parameters in the previous section, indicate that reducing the information contained in the amplification curve is beneficial. This section explores how the "$c$" parameter preserves distance information from singleplex to multiple environments of each primer set/target reaction.

In the 3-plex validation, each singleplex assay was tested against its specific target ($N = 9$), resulting in 27 different combinations of simulated multiplexes. Moreover, the "$c$" parameters were fitted and extracted from 27 empirically tested multiplex assays (81 tests). Supplementary Fig. 2 shows the correlation between simulated and empirical ADS and MDS calculated from "$c$" parameters with correlation coefficients of 0.973 and 0.774, respectively. To further evaluate whether "$c$" parameter distributions were maintained in the translation to empirical multiplexes, their three distributions (where three is equal to the number of multiplexed targets) from the singleplex reaction were compared with the corresponding distributions in empirical multiplex reactions. As illustrated in Fig. 3a–c, distributions of three different multiplex assays are visualised with their relative mean values represented by the dashed/dotted lines. The figures show the capabilities of the "$c$" parameter to maintain distance information going from simulation to

## a) ADS of 3-plex experiment

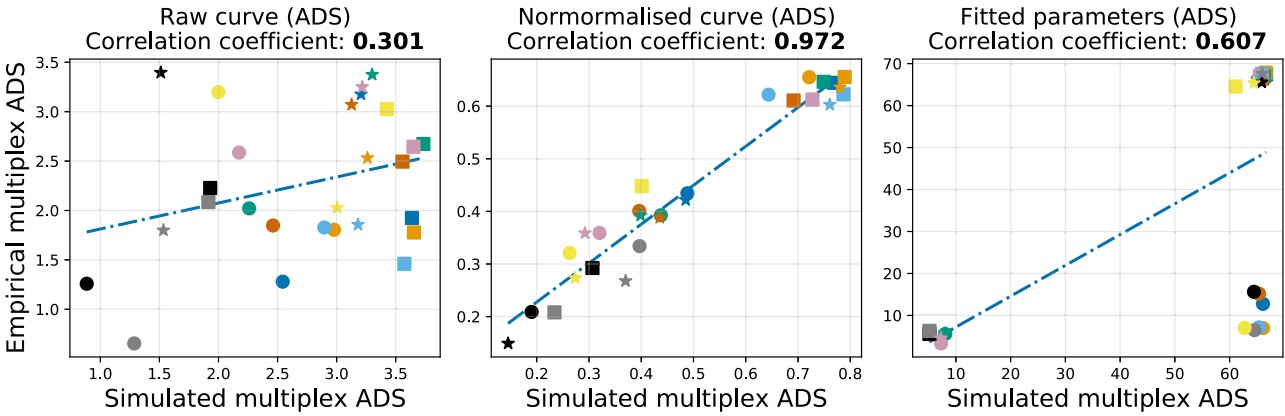

## b) MDS of 3-plex experiment

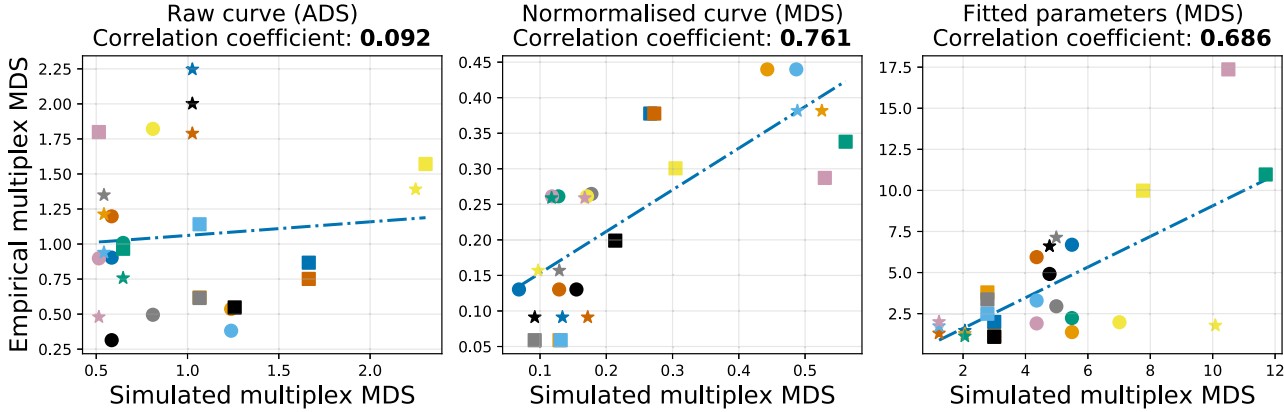

## c) Curve type of 3-plex experiment

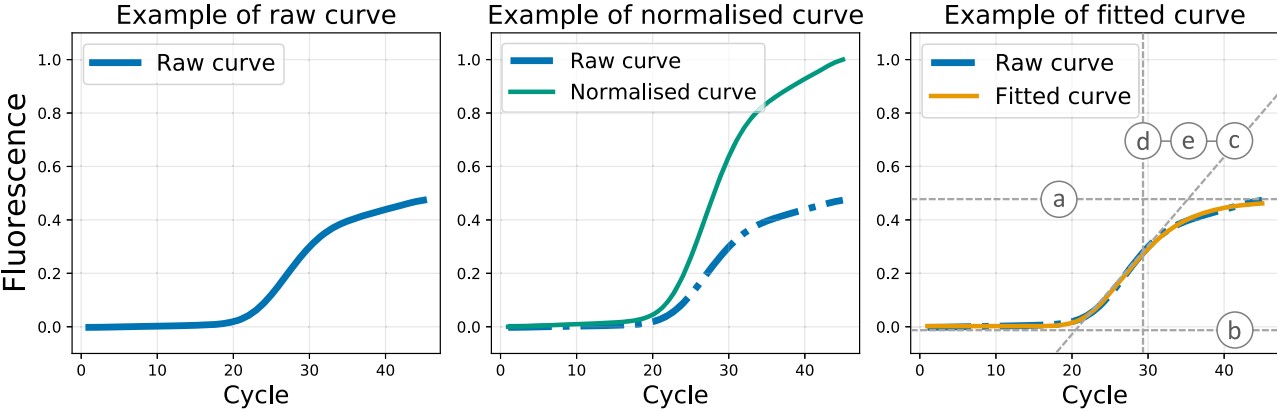

**Fig. 2 Representative features investigation based on the 3-plex assay. a** The correlations of the Average distance score (ADS) between simulated and empirical multiplexes for the three types of curves/parameters (Raw curve, normalised curve and fitted parameters) are presented (from left to right in the same order). For each plot, each point with unique colour and shape corresponds to combination 1 to 27. The blue dashed lines are computed using linear regression. The Pearson coefficients for all three plots are calculated. **b** Similarly, the correlations of Minimum distance score (MDS) are depicted for the three curve representations. **c** Illustration of the three types of curve representations. Examples of raw amplification curve (after data processing), normalised curve (computed based on the FFI) and fitted curve/parameters are presented from left to right. The fitted curve is computed with a five-parameter Sigmoid function using raw curves. As a result of this, we can obtain both fitted parameters ("*a*", "*b*", "*c*", "*d*", "*e*") and fitted curve (predicted fluorescence values corresponding to each cycle from the 5-parameter Sigmoid model with fitted parameters).

empirical test. It can be observed that in most cases, the location of the parameter distribution for each target is maintained. In other situations, the distribution may be shifted from the singleplex events; however, the relative distance relationship of

"*c*" values is retained. Fig. 3a illustrates the "*c*" parameter distribution of a low-rank ADS/MDS multiplex, showing overlaps for all three singleplex assays in both simulated and empirical multiplexes. As distances among amplification curve shapes can

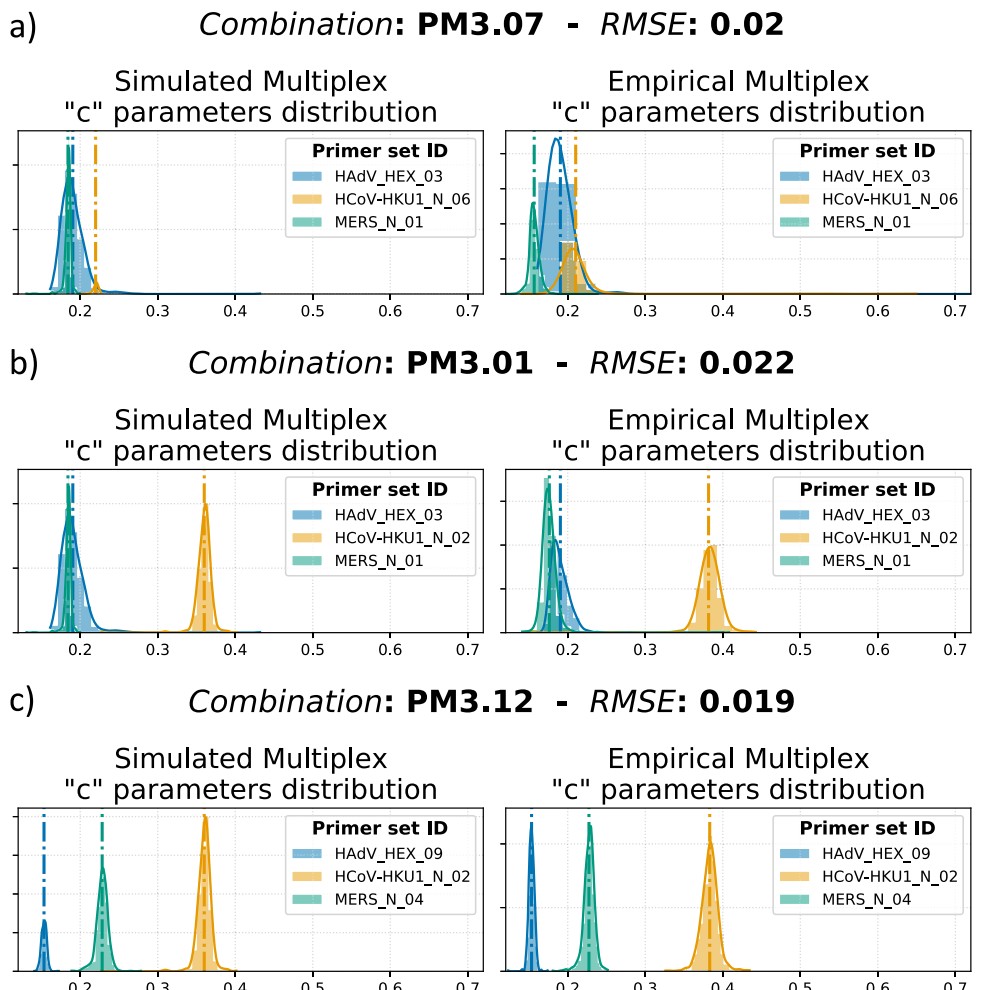

**Fig. 3 Relative "c" parameter distributions of three different multiplex assays. a** Primer Mix 3.07 (PM3.07) illustrates the "c" parameter distribution of a low-rank ADS/MDS multiplex. **b** PM3.01 as an example of high ADS but low MDS multiplexes. **c** Multiplex assay with high ADS and MDS with clearly separated distributions. For each subplot, the left graph shows the distributions of "c" parameters for the Simulated Multiplex. The right plot represents the corresponding distributions according to the empirical multiplex data. The vertical dashed lines correspond to the mean of the distribution computed for different targets. To quantitatively verify that the distances are maintained in the transition from simulated to empirical multiplexes, the RMSE of distances is calculated and displayed on the graph title. Additionally, to facilitate visualisation of this distribution plot, Supplementary Fig. 3 displays the distribution of "c" of the simulated and empirical multiplex for each target.

significantly affect the ACA classifier, reduced performance is expected for multi-target identification. Another distribution trend among multiplex assays is represented in Fig. 3b, where the selected Primer Mix (PM3.01) has a high simulated ADS value (0.117) but low MDS (0.003). Moreover, we reported that the ADS value for distributions in Fig. 3c equals 0.138, which differs only 0.21 from the combination PM3.01. However, PM3.12 has an MDS value of 0.075, representing an increase of 0.072 compared to PM3.01. This highlights the importance of considering minimum distances between "c" parameter distributions of the two closest targets: a small MDS value indicates a less separable group of target clusters, resulting in low ACA accuracies for multi-pathogen identification in a single fluorescent channel reaction. To numerically report how distributions are related in the translation from simulated to empirical multiplexes, we calculated the rooted mean squared error (RMSE) as follows:

$$\text{RMSE} = \sqrt{\frac{(D_s - D_m)^T (D_s - D_m)}{N_d}} \qquad (3)$$

where $D_s$ and $D_m$ are vectors for distances among targets in

singleplex and multiplex, respectively. The computed RMSE values for all the 3-plex combinations ranged from 0.003 to 0.050, which are negligible in comparison to the range of the "c" parameters. Given that the "c" parameter distributions in our study exhibit minimal noise and little variation in shape, and considering that our primary focus was on assessing relative distance rather than distribution similarity, RMSE proved to be a reliable criterion for evaluating the differences between distributions. The ADS, MDS, and RMSE values for all the 3-plex combinations are reported in Supplementary Table 5. These results emphasise that distances between simulated and empirical multiplex share high similarity across different ranks, ensuring that our scoring system (based on ADS and MDS) is not affected whether in singleplex or multiplex environments.

**Accuracy of all the possible combinations in 3-plex assays**. One of the aims of the Smart-Plexer is to improve the classification of multiplex assays, in our case, related to the ACA method. As demonstrated in the previous section, distances among amplification curves of empirical multiplex assays are similar to those generated in simulated multiplexes. Therefore, leveraging ADS

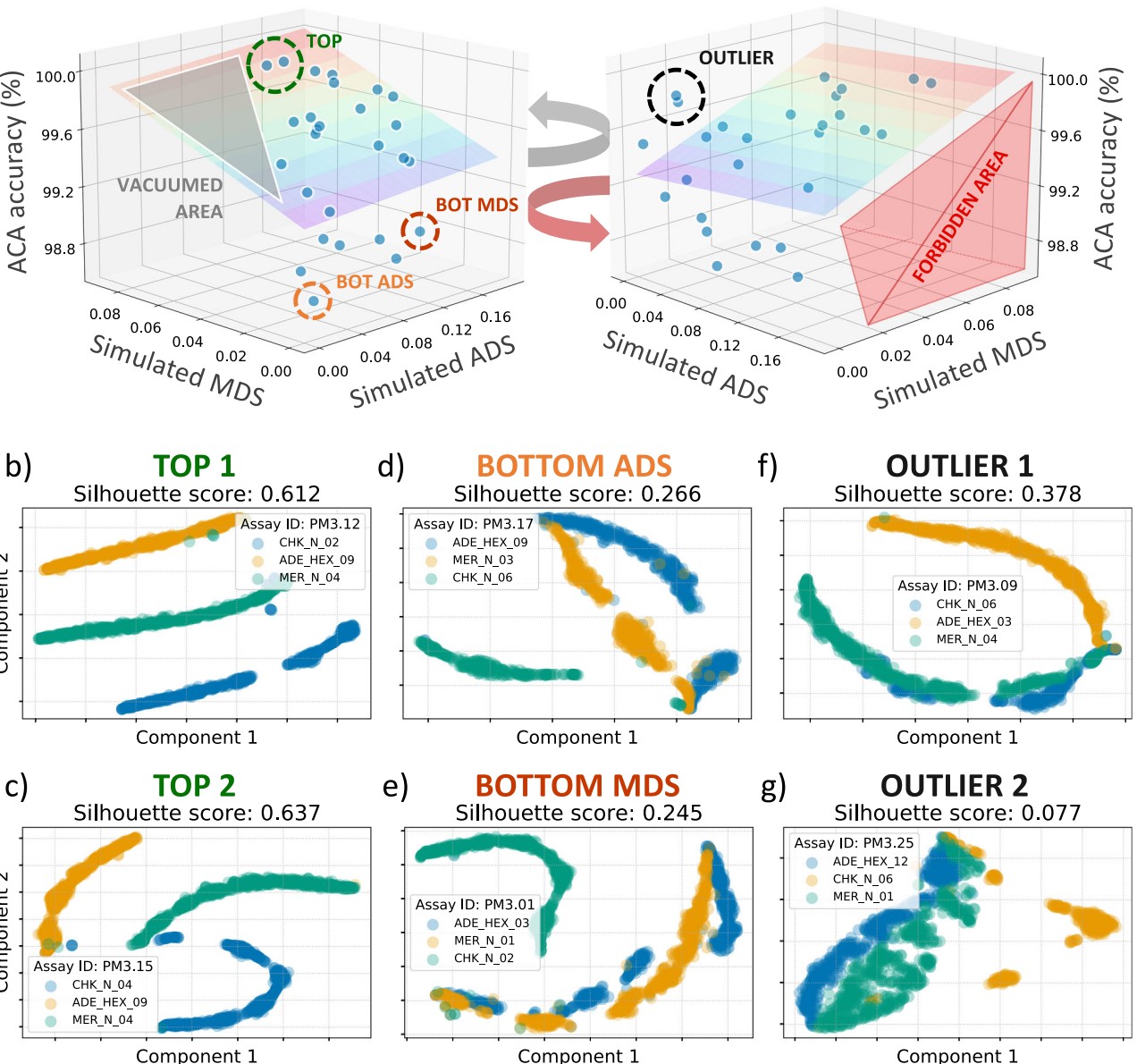

**Fig. 4 The influence of ADS/MDS on the ACA performance for all possible 3-plex combinations. a** 3-D plot of ACA classification accuracy for each combination versus simulated ADS and simulated MDS computed based on the "*c*" parameter. The rainbow plane is calculated using linear regression. In the left 3-D figure, the grey highlighted area is called the vacuumed area, where simulated MDS is larger than simulated ADS (combinations in this area are mathematically impossible to be found). The right 3-D figure is a rotation of the left one, where a red is highlighted named Forbidden Area. In this region, high ADS/MDS combinations possess low ACA accuracies; however, no combinations were found. **b–g** For the combination circled (TOP, BOT MDS, BOT ADS, and OUTLIER) in **a**, 2-D t-SNE was applied on raw curves. In addition, for quantitative verification, the mean Silhouette scores (MSS) of target clusters were reported in the subplot title.

and MDS, simulated multiplexes can be used to rank each combination and find the optimal assays with the largest inter-target distances for the ACA classifier. To further demonstrate that the ADS and MDS are crucial to improving multi-target identification in single well PCR reactions, we assess the classification performance of the ACA method by using 10-fold cross-validation and the k-Nearest Neighbours (k-NN) algorithm. Fig. 4a shows a 3-D graph where both ADS and MDS of the "*c*" parameters are correlated to the ACA accuracy. Accuracy percentages ranged from 98.63% to 100% for each multiplex. The rainbow plane, which is fitted with linear regression on all the visualised data points, represents the gradient of the classification

accuracy, showing an upward trend as ADS and MDS increase, which is consistent with our hypothesis that the ACA classification performs better with larger inter-target distances. Moreover, the plane on the left of Fig. 4a has a grey highlight zone called vacuumed area, where data points cannot fall inside as it is mathematically impossible to have an average distance value smaller than the minimum distance. We also defined another area called forbidden area, as visualised in the rotated 3-D plot on the right of Fig. 4a, where it is expected that no point will be founded, provided high values for ADS and MDS.

Both 3-D plots have circled points labelled as the top combination (TOP), bottom combination with lowest ADS

(BOT ADS), bottom combination with lowest MDS (BOT MDS), and outlier combination (OUTLIER), with ACA classification accuracies of 99.9%, 99.89%, 98.06%, 99.01%, 99.82%, and 99.87%, respectively. Although the overall classification performance for all 27 combinations shows a high average of 99.51% ± 0.41%, an increase of 1.84% is observed for the top ADS/MDS data point compared to the bottom one. Furthermore, as depicted in Fig. 4b–e, by applying 2-D t-distributed stochastic neighbour embedding (t-SNE)[43] visualisation on curves generated by the top and bottom-ranked primer combinations, more condensed target clusters and better separated inter-target boundaries can be seen for top-ranked assays. This results in more distinguishable curve shapes and larger curve distances among targets, which benefits the ACA classification. Numerical analysis of the visualised clusters was assessed using the mean silhouette scores (MSS). As reported by Kaufman et al. 2009, Silhouette scores between 0.51 and 0.70 are considered more effective in cluster separation than values below 0.50[44]. The reported MMS scores show significantly larger inter-cluster distances for the top combinations, with values >0.61 as opposed to the bottom ones of <0.27 (in Supplementary Table 5, we also report ADS, MDS, MSS, and ACA accuracies for each combination of the 3-plex experiment). This finding proves that the ADS and MDS metrics are valid indicators for predicting optimal primer set combinations for the ACA classifier. Relying on the Smart-Plexer for selecting multiplex assays from singleplexes, the likelihood of accurate multi-target identification in a single fluorescent channel reaction is significantly increased using the ACA methodology.

As mentioned above, Fig. 4a highlights the presence of outlier combinations where small ADS/MDS with high ACA accuracy are reported (instead, low accuracy for the ACA classifier is expected). However, the existence of such data points does not deny the effectiveness of the proposed method. It is important to emphasise that the overall ACA accuracy for 3-plexes is inherently high because of the low levels of multiplexing. Classifying three different curve shapes does not represent a major challenge for this Machine Learning method, and targets with minor curve-shape differences can be easily separated in the feature space. Considering this, along with the prevalent randomness that exists in the ACA method for 3-plex, accuracies higher and lower than expected may occur in the given dataset. In fact, in the area with low ADS/MDS, we can observe a large standard deviation for accuracies among data points that fall beneath and above the fitted plane. Regardless of the accidentally high accuracies and low ADS/MDS caused by randomness, Fig. 4f, g evidence that these outlier combinations will face more challenges when used for multi-target identification in larger scale multiplexes (or high-level multiplexing). In the outliers, the mapped target clusters are largely overlapped with unclear boundaries and small MSS even in 3-plex assays. Therefore, we will demonstrate in the next section that the higher the level of multiplexing is, the more difficult the target separations are in the feature space when using these outliers.

Although low ADS/MDS combinations may occasionally show good performances, the proposed method ensures that all predicted optimal multiplex assays with high ADS/MDS show high accuracies in ACA and never the opposite. As illustrated in the 3-D plots of Fig. 4a, the forbidden area (the red triangular prism) has no data point falling in, which highlights the effectiveness of the ADS/MDS ranking system. This is a first-ever demonstration that multiplex assays tailored to the ACA method can be in-silico developed starting from singleplex PCR reactions. This not only increases the likelihood of accurate multi-pathogen identification but also allows for a higher level of multiplexing in a single fluorescent channel. To demonstrate the

capabilities of the Smart-Plexer in developing optimal high-level multiplex assays for data-driven approaches, in the following section, we assess its performance with seven different targets.

**Smart-Plexer for development of 7-plex assays.** In the previous section, our focus was on using a small number of targets to demonstrate that the developed ADS and MDS used to correlate distances between curves in both simulated and empirical multiplex assays were maintained. Moreover, accuracies among all the different combinations were evaluated using the ACA methodology, where high ADS/MDS multiplex assays show the highest likelihood of correct multi-target classification. These previous results indicate that the Smart-Plexer is a promising technique for optimal selection of primer set combinations in data-driven multiplexing.

Next, we evaluated the Smart-Plexer to develop an optimal 7-plex assay, which through the ACA method, is able to accurately identify the following Respiratory Tract Infection (RTI) pathogens in a single fluorescent channel using qdPCR: Human adenovirus (HAdV), Human coronavirus OC43 (HCoV-OC43), Human coronavirus HKU1 (HCoV-HKU1), Human coronavirus 229E (HCoV-229E), Human coronavirus NL63 (HCoV-NL63), Middle East respiratory syndrome-related coronavirus (MERS-CoV), and Severe acute respiratory syndrome coronavirus 2 (SARS-CoV-2). We designed at least two different assays for each target, for a total of 24 singleplexes across the seven pathogens, as shown in Supplementary Table 6. Each primer set was tested using the synthetic DNA of its correspondent pathogenic target. Following the previous 3-plex experimental workflow, the resulting raw curves were processed, fitted, and passed to the Smart-Plexer to calculate all possible 7-plex combinations ($N = 4608$) and compute their ADS/MDS. Based on "$c$" parameter distances from fitted simulated multiplexes, Fig. 5a shows how the ADS and MDS can be visualised in a two-dimensional space. By considering the mean and standard deviation of the two scores, we set up boundaries to the ADS/MDS distribution for all the combinations and divided the space into four separate regions, with the purpose of showing how empirical multiplexes would perform for the ACA method depending on their ADS/MDS. The black horizontal segmented line in Fig. 5a divides high and low MDS, and the vertical one separates the two ADS regions, resulting in four distinct areas. By testing different multiplexes from each of these regions, we are aiming to further demonstrate that chance of developing a reliable multiplex can vary based on the selected regions or selection criteria. Therefore, we chose multiplex assays from different areas and categorised them into five classes, which were empirically tested with synthetic DNA in qdPCR: BOT ($N = 6$), MID ($N = 6$), BEST ($N = 6$), TOP-ADS and TOP-MDS ($N = 6$) values (detailed selection criteria are reported in the "Methods" section).

After the empirical testing, the distances of the "$c$" parameters of each selected multiplex were compared to the simulated one, resulting in a correlation coefficient of 0.99, as shown in the middle graph of Fig. 5b. Moreover, empirical multiplex amplification events were visualised using 3-D t-SNE, and distances across target clusters were calculated with the MSS. As shown in the left plot of Fig. 5b, clusters of the selected BOT combination have an MSS of 0.12, whereas for the BEST one, the score is 0.67. It can be observed that there is a clear difference in clustering between the two selected multiplex assays, where the BEST one shows clear separation among different targets (in line with the 3-plex results), and is expected to converge in better ACA classification. The opposite scenario is shown in the BOT combination.

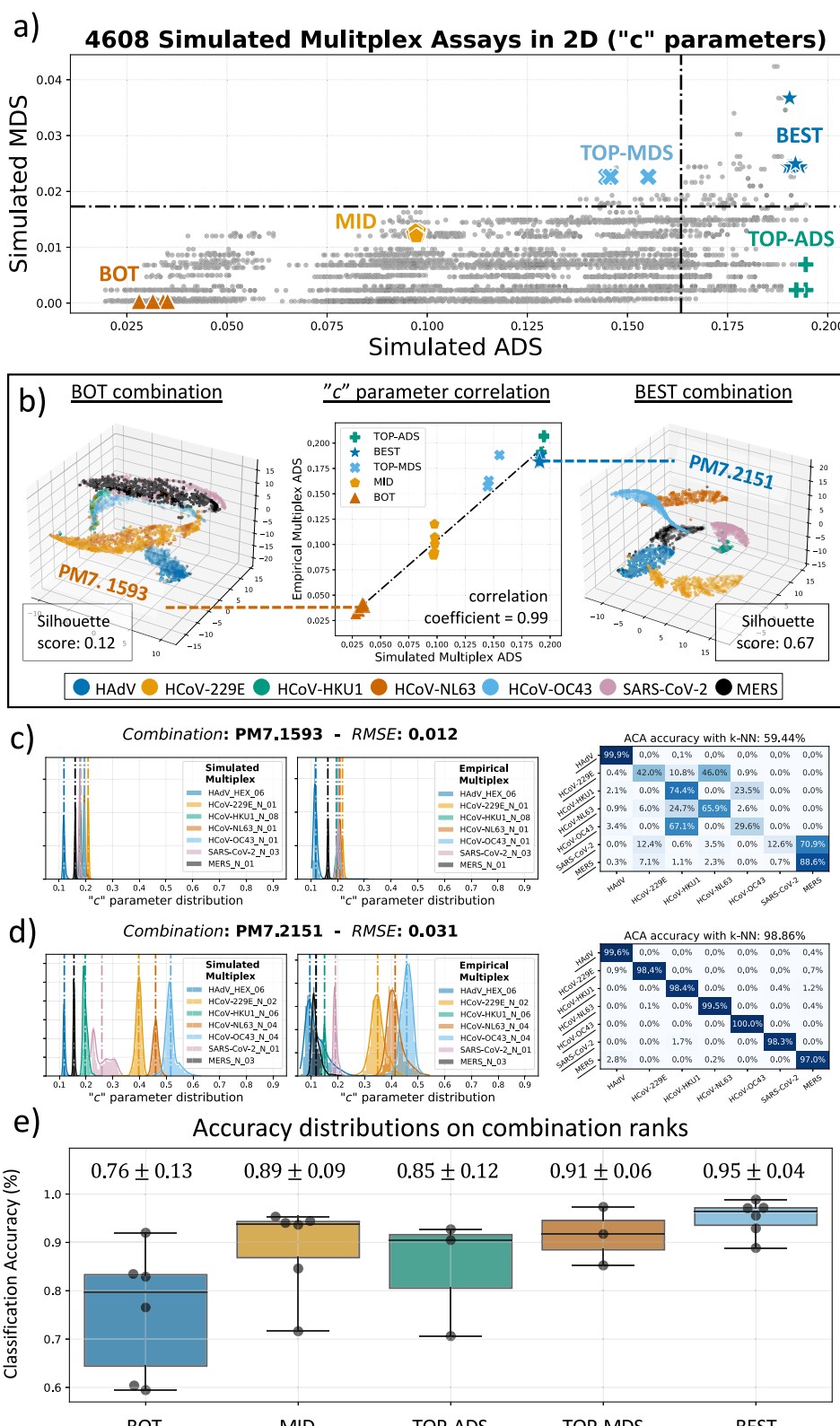

Higher level multiplexing, distance distributions of the "$c$" parameters were still maintained from simulated to empirical testing; therefore, we computed the *RMSE* of the chosen tested combinations. Fig. 5c, d illustrates side-by-side "$c$" parameter distributions for each target in both simulated (left) and empirical (right) multiplexes, showing a small RMSE for both BOT and BEST assays (0.012 and 0.031), and confirming the distance-

maintaining hypothesis validated in the 3-plex experiments. The ACA accuracy was evaluated using training and testing datasets obtained in different experimental settings (different days, operators, and reagents) to assess the methods reproducibility. As expected, the performance of the BEST combination was significantly higher than the BOT one, with a 39.42% increase in accuracy. Furthermore, in Supplementary Table 7, we reported

**Fig. 5 Validation of Smart-Plexer based on 7-plex assays. a** 2-D ranking results for all 4608 combinations in 7-plex based on simulated ADS and simulated MDS. The plot is divided into four regions to explore the relationship between ACA performance and ADS/MDS. Combinations from five different classes (BOT, MID, BEST, TOP-ADS, TOP-MDS) were selected for multiplex empirical testing. **b** The 2-D plot in the middle depicts the relationship between empirical and simulated scores based on "c" parameters. Enlarged data points for one of the BOT (PM7.1593) and BEST (PM7.2151) combinations are visualised with 3-D t-SNE on raw curves, and the corresponding Silhouette scores are calculated. **c** and **d** Simulated and Empirical "c" distributions of the selected combinations (PM7.1593 and PM7.2151) are plotted (RMSE values in subplot titles). The vertical dashed lines correspond to the mean of the distribution computed for different targets. On the right, the confusion matrixes of ACA performance for both cases are presented, and overall accuracy using k-NN is reported in the title. True labels are on the y-axis and ACA-predicted labels are on the x-axis (each target sensitivity is also reported in percentage). **e** The box plot of ACA classification accuracy for each selected group. The mean and standard deviation of ACA accuracy on empirical multiplexes are calculated and shown on each box bar.

the ADS, MDS, and accuracy values for the 24 selected multiplex assays. In Supplementary Fig. 4, we also visualised the standard curve for each target using the BEST 7-plex assay to evaluate primer sensitivity and specificity. The chosen multiplex reached a limit of quantification equal to $10^2$ for all the respiratory pathogens using synthetic DNA in real-time PCR.

As described before, ACA performances were evaluated using training and testing datasets from different experimental settings with the same sample size. All the selected 24 multiplexes were empirically tested, and their multi-target identification performances were assessed. In Fig. 5e, accuracies and standard deviations of each group of multiplexes were reported and visualised as box plots. The best-combination group scored an average (±standard deviation) classification performance of 95% (±0.04%) using a k-NN classifier, which is the highest average and the lowest standard deviation among all the groups. There is a decreasing trend in the average accuracy, and an increasing trend in the standard deviation as the ADS/MDS values become smaller. Previously, the 3-plex validation showed the presence of outliers in low ADS/MDS rank with high ACA classification accuracy, which is also observed in these 7-plex tests. However, the standard deviation indicates that the Smart-Plexer does provide a robust and solid solution (even at high-level multiplexing) to significantly increase the likelihood of choosing an optimal multiplex for data-driven multiplexing (i.e. ACA methodology).

**Validation with clinical isolates**. The developed 7-plex was evaluated on clinical isolates for multi-pathogen identification. After testing six potential best combinations based on ADS/MDS, the multiplex with the highest ACA classification accuracy on synthetic DNA (PM7.2151) was selected. Inactivated clinical samples were purchased from Randox Laboratories (UK) and nucleic acid was extracted using the QIAGEN mini am kit, according to the manufacturer's instructions. The extracted samples were used as the testing dataset (7638 positive amplification reactions), while curves resulting from synthetic DNA amplification reactions (5207 positive amplification reactions) formed the training set. The classifier used was a k-NN with a number of neighbours equal to 10. As shown in Table 1, a total of 14 positive samples were classified in qdPCR using the ACA methodology. The predicated label of a sample is given by selecting the most predictable label within all the in-sample curves. The confidence level was given as the percentage of the amplification curves with the most predicted label. All pathogens using the Smart-Plexer selected candidate assay, all pathogens were currently identified with high confidence (median = 95.46%).

It is important to note that this study faced a seven-class classification problem, where the accuracy of a "random guess" (or a random classifier as convention) equals 14.3% under a balanced dataset. All the confidence levels were much higher than the random guess accuracy, indicating solid and robust

predictions with the selected optimal multiplex assay. Although the number of clinical samples was limited by the number of pathogens provided by the manufacturer, the proposed framework, in combination with the ACA methodology, achieved a highly accurate identification of multiple pathogens by using an optimal multiplex assay in a single fluorescent channel reaction. The Smart-Plexer can leverage the capability of data-driven multiplexing to an easy-to-develop, robust, and cost-effective molecular diagnostic solution.

## Discussion

In this work, we developed the Smart-Plexer, an innovative framework that combines wet-lab experiments and computational algorithms to generate optimal multiplex assays for data-driven approaches using real-time PCR data. The method leverages mathematical metrics to construct an advanced ranking system to increase the throughput of conventional molecular tests by optimising their chemical peculiarities. To reveal the potential of this powerful approach, we demonstrated it with a recently reported machine learning method, named Amplification Curve Analysis (ACA), which is capable of identifying multiple nucleic acid targets in a single fluorescent channel with conventional PCR instruments. As the ACA leverages kinetic information encoded in the amplification curve, multiple targets can be classified based on the unique interaction with their assigned primer sets. However, constructing different amplification curve shapes for each multiplexed target is one of the major challenges for the ACA approach. The Smart-Plexer solves this problem by providing an easy-to-use framework for multiplex assay development, enabling high-level and highly accurate data-driven multiplexing.

This study describes the development of the Smart-Plexer. The workflow was initially evaluated on the 3-plex panel. From the wet-lab testing of three singleplex assays for each of the three targets, a total of 27 combinations (in our case 3-plex assays) can be generated in silico (simulated multiplex) and ranked based on the mathematical curve-shape distances. Using synthetic DNA in qdPCR and a single fluorescent channel, the assays were empirically tested (empirical multiplex), and the ACA classification accuracies were evaluated for all the possible combinations. The distance scores computed from the Smart-Plexer for multiplex assay ranking were linearly correlated between simulated and empirical multiplexes and between high-rank multiplexes. The smart-plexer increased ACA accuracies, confirming that the metrics used in this novel framework are theoretically connected to the distance measurement of the machine learning classifier.

The design and utility of the Smart-Plexer were assessed for a 7-plex assay targeting common respiratory tract infection (RTI) pathogens. Consistent with the 3-plex validation, the correlation between simulated and empirical multiplex was maintained in 7-plex. Regarding the ACA classification, it was logical that higher similarities among curves existed in a scenario with a higher number of targets, making it harder to develop multiplex assays. Nevertheless, the Smart-Plexer generated an optimal multiplex

**Table 1 Validation with clinical isolates.**

| Sample index | Panel ID (Randox, UK) | Expected pathogen (true label) | ACA classified pathogen (predicted label) | AC count | Confidence level (%) | Outcome |
|---|---|---|---|---|---|---|
| 1 | QAV164189 | HAdV | HAdV | 14 | 100.0 | Detected |
| 2 | QAV164189 | HCoV-NL63 | HCoV-NL63 | 770 | 100.0 | Detected |
| 3 | QAV164189 | HCoV-NL63 | HCoV-NL63 | 545 | 96.15 | Detected |
| 4 | QAV164189 | HCoV-OC43 | HCoV-OC43 | 94 | 78.72 | Detected |
| 5 | SCV2QC | SARS-COV-2 | SARS-COV-2 | 769 | 69.96 | Detected |
| 6 | SCV2QC | SARS-COV-2 | SARS-COV-2 | 631 | 94.77 | Detected |
| 7 | SCV2QC | SARS-COV-2 | SARS-COV-2 | 766 | 100.0 | Detected |
| 8 | SCV2QC | SARS-COV-2 | SARS-COV-2 | 756 | 99.34 | Detected |
| 9 | SCV2QC | SARS-COV-2 | SARS-COV-2 | 748 | 99.20 | Detected |
| 10 | QAV154181 | MERS | MERS | 287 | 60.98 | Detected |
| 11 | QAV154181 | MERS | MERS | 770 | 96.49 | Detected |
| 12 | QAV154181 | MERS | MERS | 770 | 79.09 | Detected |
| 13 | QAV154181 | MERS | MERS | 698 | 91.69 | Detected |
| 14 | QAV154181 | MERS | MERS | 20 | 70.00 | Detected |

assay, which correctly identified pathogens presented in 14 commercial clinical samples. It was further demonstrated that, since ACA is a clustering method, it required a large minimum distance between the two closest clusters and a large average distance among all clusters in the multiplex. Therefore, the Smart-Plexer ranking system enabled the development of optimal multiplex assays for data-driven multiplexing.

Apart from the scalability of multiplexing that the Smart-Plexer can provide to the ACA method, we demonstrated for the first time that machine learning approaches can be applied to probe-based multiplexes, in our case, TaqMan. Probe-based assays, together with the use of intercalating dyes and isothermal chemistries, are expanding the boundaries of data-driven multiplexing and opening new windows for its application in commercial, research and clinical fields. The Smart-Plexer eases the development of any novel multiplex panel or molecular assays, enabling the use of the ACA as an emerging diagnostic tool. Through this hybrid method, it is possible to select the highest rank combination in silico with wet-lab tested singleplex, avoiding performing expensive and time-consuming multiplex assay development phases.

While this novel framework is validated with high-level multiplexing (7-plex), it is essential to highlight that distances between amplification curves can be a limiting factor in single fluorescent channel multiplexing. This affects the Smart-Plexer since the inter-target differences of fitting parameters considered for the distance measurement become smaller as the target number increases. In this work, we use linear distance measurements, but more advanced metrics (e.g. Minkowski, Chebyshev or Cosine) can be adopted to improve the ranking performance. Moreover, when a higher level of multiplexing is required, the use of probe-based chemistries such as TaqMan is ideal as it reduces the number of non-specific detection and enables the use of multiple fluorescent channels. By leveraging the optical capability of real-time PCR instruments, a multiplex assay using multiple-channel detection can double or triple the number of targets in a single reaction[45–47]. All these strategies aim to improve the ACA classification through a more innovative development from the chemistry perspective, while from the machine learning view, the current classifiers rely on state-of-the-art algorithms which shine for their robustness but are limited for tailoring to specific datasets. We previously demonstrated that more advanced classifiers, such as convolutional neural networks (CNN), have the potential to enhance the ACA's capability for classifying targets in higher-level multiplex assays. Additionally, efforts have been made to improve target classification using advanced frameworks like the Transformer-based Conditional Domain Adversarial Network (T-CDAN), as described by Mao et al. in 2023, to address the issue of domain discrepancy in amplification curve analysis[48]. However, as a novel technique, data-driven multiplexing requires further optimisation and algorithm development to maximise its potential.

The Smart-Plexer provides a solution to develop multiplex assays using empirical testing and *in-silico* computation. The requirement for some wet-lab experiments can limit its application in terms of staff training and time requirements. To further validate the practical application of the Smart-Plexer method, future work using different platform (such as qPCR) and larger clinical samples cohort is currently in progress to ensure that the Smart-Plexer meets the requirements of high-throughput multiplexing, and sensitive detection methods in clinical diagnostics. By continuously refining and optimising the methodology, we aim to establish a practical platform that offers cost-effective, rapid, and accurate detection of pathogenic microbial infections and other nucleic acid multiple detection needs.

Moreover, our future work will focus on the full automation of developing such assays. Novel methodologies to predict amplification curve behaviours will be developed. One example is the brand-new algorithm for designing multiplex PCR primers using dimer likelihood estimation by Xie et al.[21] Another future aspect of this research is to increase inter-target curve shape differences further. An example would be in probe-based chemistries, where modifying amplification curve shapes can be achieved by changing the concentration levels of the fluorescent probe. In this way, we can expand inter-target distances of amplification curves to ease the ACA classification with better clustering performance. This work will broaden the application of the ACA method/and significantly increase its flexibility and scalability.

In this work, we present for the first time a complete pipeline, Smart-Plexer, for developing optimal multiplex assays and opening the use of the ACA method to the broad scientific community. Smart-Plexer will support optimal multiplex design to improve the accuracy and cost of molecular diagnostics.

## Methods

**Synthetic double-stranded DNA templates**. Double-stranded synthetic DNA was used in this study to develop and assess the performance of all singleplex assays. In particular, we used the entire coding sequence of the hexon protein gene (HEX gene) for

human adenovirus (HAdV), and the nucleocapsid protein gene (N gene) of human coronavirus OC43 (HCoV-OC43), HKU1 (HCoV-HKU1), 229E (HCoV-229E), NL63 (HCoV-NL63), Middle East respiratory syndrome-related coronavirus (MERS-CoV) and severe acute respiratory syndrome coronavirus 2 (SARS-CoV-2). The following NCBI accession numbers were used as references for the gBlock synthesis: NC_001405, NC_006213, NC_006577, NC_002645, NC_005831, NC_019843 and NC_045512, respectively. The synthetic constructs were used for qPCR experiments when determining the limit-of-quantification of each PCR assay and in qdPCR experiments for generating the dataset used in the simulation of the multiplexes and their empirical testing. The gene fragments (ranging from 1134 to 1558 bp) were purchased from Integrated DNA Technologies Ltd. (IDT) and resuspended in Tris−EDTA buffer to 10 ng/μl stock solutions (stored at −80 °C until further use). The concentrations of all DNA stock solutions were determined using a Qubit 3.0 fluorimeter (Life Technologies).

**Clinical isolates.** Whole pathogen control panels were purchased from Randox Laboratories Ltd, including MERS-CoV (catalogue no. QAV154181), CoV-OC43, NL63 (catalogue no. QAV164189), and SARS-CoV-2 (catalogue no. SCV2QC). Samples were extracted using the QIAamp Viral RNA Mini Kits (catalogue no. 52906). Viral nucleic acid was extracted using the manufacturer-recommended protocol[49]. Viral RNA was reverse transcribed to cDNA using Fluidigm reverse transcription master mix (catalogue no. SKU 100–6299). Viral cDNA was further pre-amplified using Fluidigm Preamp master mix (catalogue no. PN 100-5744). Reverse transcription and pre-amplification were conducted according to the Fluidigm manufacturer's protocol (Fluidigm document number: 101-7571 A2 and 100-5876 C2).

**PCR assay design.** The sequences of each gene were downloaded from the GenBank website[50]. Based on the comprehensive analyses and alignments of each type using the MUSCLE algorithm[51], primers were specifically designed to amplify all sequence variations within each gene belonging to their specific target (inclusivity) and to exclude closely related but not inclusive sequences (exclusivity). Design and in-silico analysis were conducted using GENEious Prime 2022.0.1[52]. Primer characteristics were analysed through IDT OligoAnalyzer software[53] using the J. SantaLucia thermodynamic table for melting temperature ($T_m$) evaluation, hairpin, self-dimer, and cross-primer formation[54]. To confirm the specificity of the real-time digital PCR assays, the primers were first evaluated in a singleplex PCR environment to address their specificity and sensitivity for both singleplex and multiplex assays. All primers were synthesised by IDT (Coralville, IA, United States). Details on primer sequences, along with multiplex assay combinations, are provided in Supplementary Tables 8, 9 (for both 3-plex assays), and Supplementary Tables 10, 11 (for both 7-plex assays).

**Real-time digital PCR.** For real-time amplification experiments, we used the BioMark HD (Fluidigm) and the QIAquant 96 5plex (catalogue no. 9003011). The master mix used was the Prime-Time Gene Expression Master Mix from Integrated DNA Technologies (IDT, catalogue no. 1055772) supplemented with ROX passive reference dye and pre-mixed following manufacturer guidelines. The qdPCR was performed with Fluidigm qdPCR 37k integrated fluidic circuits (IFC) (catalogue no. SKU100-6152) and was supplemented with Fluidigm 20X GE loading buffer (PN 85000746). The priming and loading steps of the IFC were

followed as per the supplier's protocol (Fluidigm document number: 100-6896 Rev 03). Each amplification mix for the qdPCR experiment contained 3 μl 2X IDT PrimeTime Gene Expression Master Mix (with passive ROX), 0.6 μl 20X GE, 0.6 μl 10X Primer mixture, 1.8 μl DNA templates from synthetic DNA, pre-amplified cDNA, or controls, and to bring the final volume to 6 μl. A total of 4.5 μl of reaction mix was transferred to each inlet (or panel) of a Fluidigm 37k IFC for the thermal cycling step. Thermal-cycle conditions consisted of a hot start step for 3 min at 95 °C, followed by 45 cycles at 95 °C for 15 sec and 60 °C for 45 s. Real-time data of the amplification events were exported as a text file for each bulk by Fluidigm Digital PCR Analysis software (version 4.1.2).

**Limit-of-quantification.** We used real-time PCR from QIAGEN (QIAquanta96) to evaluate the Limit-of-quantification (LoQ) of the selected 7-plex assay. Standard curves were generated with synthetic DNA ranging from $10^7$ to $10^1$, apart from SARS-CoV-2 whose concentration was from $10^5$ to $10^1$ because of limitations due to pandemic suppliers (IDT). PCR data were extracted and processed according to the data processing step. Standard curve plots and statistical values are reported in Supplementary Fig. 4. The Absence of amplification signals was detected in negative template control (NTC).

**Data processing.** The processing of raw amplification curves is comprised of three parts. Firstly, to ensure all curves start from approximately zero fluorescence value and to normalise the starting cycles of the curve across the entire time series, the background information was removed, which can be expressed as

$$\text{Fl}_{br}(t) = \text{Fl}(t) - \text{avg}_{back} \qquad (4)$$

where $\text{Fl}_{br}(t)$ represents a curve with the background removed and $\text{Fl}(t)$ is the raw fluorescence values for each cycle $t = 1, 2, \cdots, T$. Here $T$ indicates the total number of cycles for each amplification curve (45 in our case), and $\text{avg}_{back}$ is the average background value. In order to avoid instrumental noise commonly found at the beginning of the PCR reaction, the $\text{avg}_{back}$ value was estimated as the average value of the first several cycles' fluorescence, excluding the initial ones. In our case, five cycles were considered for the flat phase and the first three cycles were skipped. Secondly, late amplification filtering was applied to select curves that reached the plateau phase. The basic idea is to estimate the cycle threshold value ($\text{Est}_{Ct}$) for each curve, which can be represented as

$$\text{Est}_{Ct} = \min t \qquad (5)$$

$$\text{s.t.} \; \frac{\text{Fl}_{br}(t) - F_{min}}{F_{max} - F_{min}} \geq F_{th} \qquad (6)$$

where $t \in \{1, 2, \cdots, T\}$, and $F_{max}$ and $F_{min}$ represent maximum and minimum fluorescence values of the entire reaction respectively for each curve. $F_{th}$ is the fluorescence threshold and curves whose $\text{Est}_{Ct}$ are above the cycle threshold ($C_t = 30$ as suggested by the manufacturer) were removed. Lastly, a filter was applied to remove non-sigmoidal curves with excessive noisy signals. The sigmoidal trend of a noisy curve may contain certain notches. Based on this feature, we estimated the first derivative of each curve using:

$$\text{Fl}_{br}{}'(t) = \text{Fl}_{br}(t) - \text{Fl}_{br}(t-1), t = 2, \cdots, T \qquad (7)$$

The number of zero-crossing points in $\text{Fl}_{br}{}'(t)$ is related to the number of notches in the curve. Therefore, noisy curves should have significantly more zero-crossing points in their first derivatives compared with smooth sigmoidal curves. The curves

that satisfied the following condition were regarded as noisy and removed:

$$\sum_t \frac{-\text{sgn}\left[\text{Fl}_{\text{br}}'(t)\right] + 1}{2} > N_{\text{zc}} \tag{8}$$

where sgn[·] is the sign function and $N_{\text{zc}}$ is the given threshold value ($N_{\text{zc}} = 9$ in our research, following the concept depicted in Miglietta et al. 2022)[39].

**Five-parametric sigmoidal fitting**. Since amplification curves contain information such as background, plateau phase, and slope, we derived the most representative features of it using the sigmoidal equation. The chosen model in this study for curve fitting is the five-parametric sigmoid function, whose equation is given below:

$$f(t, \boldsymbol{p}) = \frac{a}{\left(1 + \exp^{-c(t-d)}\right)^e} + b \tag{9}$$

$$\boldsymbol{p} = [a, b, c, d, e]^{\text{T}} \tag{10}$$

where $t$ is the amplification cycle, $\boldsymbol{p}$ is the parameter vector, $f(t, \boldsymbol{p})$ is the fluorescence at cycle $t$. The mathematical function of these parameters and their corresponding representations in amplification curves are shown in Table 2:

To reduce optimisation iterations and unsuccessful fitting, we applied a pivot fitting on a subset of data ($\boldsymbol{D}_{\text{s}}$) to evaluate the optimal initial parameters $\boldsymbol{p}_0^{\text{opt}}$ for the equation before searching on the entire dataset ($\boldsymbol{D}$). First, we defined a non-linear Least Square function LS($\boldsymbol{p}$), whose equation is shown below:

$$\text{LS}(\boldsymbol{p}) = \sum_{t=1}^{T} \left(f(t, \boldsymbol{p}) - \text{Fl}_{\text{br}}(t)\right)^2 \tag{11}$$

To apply the pivot fitting, we first initialised $\boldsymbol{p_0} = [0, 0, 0, 0, 0]^{\text{T}}$. Then, for the $i$th curve $\text{Fl}_{\text{br}}^i$ within the dataset $\boldsymbol{D}_{\text{s}}$, the following optimisation problem was solved to find the fitted parameter vector:

$$\boldsymbol{p}_i = \underset{B_{\text{low}} < \boldsymbol{p} < B_{\text{up}}}{\text{argmin}} \text{LS}(\boldsymbol{p}) \tag{12}$$

where the lower bound $B_{\text{low}}$ and the upper bound $B_{\text{up}}$ for all the parameters are −100 and 100, respectively. After all the curves were fitted, the mean vector of all the $\boldsymbol{p}_i$ was used as the optimal $\boldsymbol{p}_0^{\text{opt}}$.

With the outcome from the pivot fitting, we fitted all curves in $\boldsymbol{D}$ starting from $\boldsymbol{p}_0^{\text{opt}}$. In addition, to get better fitting performance, we increased the maximum number of fitting iterations (maxfev) to a sufficiently large value (1,000,000 in our case). The same $B_{\text{low}}$ and $B_{\text{up}}$ were used for the pivot fitting.

**Calculating average distance score (ADS) and minimum distance score (MDS) for multiplex assays**. There are four curve representations for calculating ADS and MDS, which are: raw curves (45-D), normalised curves (45-D), fitted parameters (5-D) and $c$ parameter (1-D). Two steps were taken before the score calculation: (i) Extract the median feature vectors of each target

for 45-D, 5-D and 1-D feature arrays. The median value was taken on each dimension, and the median feature vector with the same dimension was generated. It is assumed that the distribution of each target is Gaussian. However, outliers can affect the distribution unexpectedly. Therefore, the median value is a more robust representative compared to the average value, and $N_t$ median vectors corresponding to $N_t$ targets were constructed. (ii) Calculate Euclidean distance between each pair of targets, where given $N_t$ targets, the total number of distances $N_{\text{d}}$ is

$$\boldsymbol{p}_i = \underset{B_{\text{low}} < \boldsymbol{p} < B_{\text{up}}}{\text{argmin}} \text{LS}(\boldsymbol{p}) \tag{13}$$

$$N_d = \binom{N_t}{2} = \frac{N_t(N_t - 1)}{2} \tag{14}$$

The vector of distances for each pair of targets is defined as:

$$S_{\text{D}} = \left[d_{ij} | \text{for each } i = 2, \dots, N_t, j = 1, 2, ..i-1\right] \tag{15}$$

where $d_{ij}$ represents the Euclidean distance between extracted median vectors of target $i$ and target $j$. With the constructed distance set, the ADS and MDS were calculated as the average and the minimum value of all elements in $S_{\text{D}}$, respectively:

$$\text{ADS} = \text{mean}(S_{\text{D}}) \tag{16}$$

$$\text{MDS} = \text{min}(S_{\text{D}}) \tag{17}$$

**ACA methodology**. The Amplification Curve Analysis (ACA) methodology was developed by Moniri et al. in 2020[31]. For the first time, shapes of amplification curves from real-time PCR data were used for multiple target identification in a single fluorescent channel reaction, utilising data-driven algorithms. The ACA takes the entire amplification time series as input and uses machine learning to classify curves into different categories of targets. This approach highlights the significance of the kinetic information embedded in amplification curves. As previously reported, several classical machine learning methods (e.g. k-NN, Random Forest, Support Vector Machine) as well as deep-learning based approaches (e.g. Convolutional Neural Networks) can be applied to the time series[32]. In this article, a k-NN classifier with 10 neighbours was used for the ACA performance evaluation. To establish the ground truth, synthetic DNA targets with known identities and clinical samples with confirmed pathogen information were utilised, providing reliable references for evaluating the ACA classifier and validating the effectiveness of the Smart-Plexer framework.

**Ranking system**. The inputs of the ranking system are simulated ADS and MDS. To increase the likelihood of choosing an optimal assay for data-driven multiplexing approaches, we considered assays with the highest ADS and MDS ($S_{\text{BEST}}$) selected from the entire combination set ($S_{ALL}$). Provided the number of the best combinations to be selected as $N_{\text{BEST}}$ and the number of total combinations as $N_c$, the steps applied are described in Table 3 (Algorithm 1). The proposed Algorithm 1

| Table 2 Five sigmoidal fitted parameters. | | |
|---|---|---|
| **Parameter** | **Mathematical meaning** | **Representation in amplification curves** |
| **a** | Amplitude of the function in the $y$-axis | Affect the maximum fluorescence that the amplification curves can reach |
| **b** | Vertical shift of the function along the $y$-axis | Affect the maximum fluorescence together with parameter $a$ |
| **c** | Maximum slope of the sigmoid function | Related to the efficiency of PCR reactions |
| **d** | Horizontal shift of the function $x$-axis | Fractional cycle of the inflection point (related to $C_t$ values) |
| **e** | Richard's coefficient | Asymmetry of the sigmoidal trend |

| Table 3 Algorithm 1. |
| --- |
| **Algorithm 1** |
| 1:  Initialise $N_{BEST}$ as required, $S_{BEST} \leftarrow \varnothing$ |
| 2:  for $n_e = N_{BEST}, N_{BEST} + 1, \ldots, N_c$ do |
| 3:    $S_{BEST}^{MDS} \triangleq \{x \mid x \text{ are the top } n_e \text{ combinations in } S_{ALL} \text{ with largest MDS}\}$ |
| 4:    $S_{BEST}^{ADS} \triangleq \{x \mid x \text{ are the top } n_e \text{ combinations in } S_{ALL} \text{ with largest ADS}\}$ |
| 5:    $S_{BEST} \leftarrow (S_{BEST}^{MDS} \cap S_{BEST}^{ADS}) \cup S_{BEST}$ |
| 6:    if $\lvert S_{BEST} \rvert \geq N_{BEST}$ then |
| 7:      return $S_{BEST}$ |
| 8:    end if |
| 9:  end for |

| Table 4 Algorithm 2. |
| --- |
| **Algorithm 2** |
| 1:  Initialise $N_{MID}$ as required, $S_{MID} \leftarrow \varnothing$, $MDS_{max}$ and $ADS_{max}$ the maximum MDS and ADS among all combinations, $ADS_{bias} = MDS_{bias} \leftarrow 0.001$ |
| 2:  $R_{MDS} \triangleq \left( \frac{MDS_{max}}{2} - MDS_{bias}, \frac{MDS_{max}}{2} + MDS_{bias} \right)$ |
| 3:  $R_{ADS} \triangleq \left( \frac{ADS_{max}}{2} - ADS_{bias}, \frac{ADS_{max}}{2} + ADS_{bias} \right)$ |
| 4:  $S_{MID}^{tmp} \triangleq \{x \mid MDS_x \in R_{MDS} \text{ and } ADS_x \in R_{ADS}, \forall x \in S_{ALL}\}$ |
| 5:  $S_{MID} \leftarrow$ apply Algorithm 1 on $S_{MID}^{tmp}$ with $N_{MID}$ |
| 6:  return $S_{MID}$ |

is used to pick the best-simulated multiplexes based on the developed metrics ADS and MDS, and these assays are further tested empirically to select the optimal one for diagnostic use. Moreover, to verify the correlation of the Smart-Plexer ranking with the ACA performance, the algorithm was used to select the bottom multiplexes with the lowest ADS and MDS, by modifying steps 3 and 4, so that the smallest instead of the largest ADS and MDS are applied.

**The Smart-Plexer Workflow**. The complete workflow of utilising the Smart-Plexer in a real laboratory setting is illustrated in Fig. 1 and depicted as follows: given a number of target genes to be identified, several candidate primer sets are first in-silico designed and tested in singleplex format for each target, resulting in real-time PCR amplification curves for all the assays. The obtained data are further processed using the background, late curve, and noisy curve removal techniques mentioned in the "Data processing" section. The processed curves are then fitted with the sigmoidal function from which the "$c$" parameters are extracted. For each potential combination of primer sets, inter-target distances of "$c$" parameters from singleplex curves are calculated and function as simulated alternatives for empirical multiplex curve distances. In this way, the best candidates for multiplex assays can be selected by choosing the combinations with the most distant target clusters (represented by "$c$") in the simulation. This progress is achieved by calculating the "$c$" parameter-based ADS and MDS of each combination and finding the best ones using the ranking system mentioned above. The best candidate assays shortlisted from simulated multiplexes further go through wet-lab tests on synthetic DNA templates, and the ACA-based target identification is applied to the empirical multiplex data. The final winner assay with the highest ACA classification performance on synthetic DNA is labelled as the optimal assay, which is the final output of the entire Smart-Plexer workflow.

**3-plex validation**. Synthetic DNA of Adenovirus (HAdV), Human coronavirus HKU1 (HCoV-HKU1), and Middle East respiratory syndrome-related coronavirus (MERS-CoV) targets were selected for a 3-plex validation, and all the data were generated in real-time digital PCR (qdPCR). Three primer sets were designed as candidates for each target, resulting in 27 potential combinations of multiplex assays in total. Because of the relatively small number of candidate assays, it is possible to perform wet-lab experiments for all combinations and analyse the relationship between simulated and empirical multiplex curve distances. Simulated ADS and MDS were calculated on different levels of curve representations (raw curves, FFI-normalised curves, and fitted parameters), and their correlations with the same metrics derived from empirical multiplex data were analysed. Furthermore, the ADS and MDS of "$c$"

parameters, which are more concise indicators for inter-target curve distances, were generated and compared between simulated and empirical multiplexes. ACA performance against simulated ADS and MDS was depicted, and the t-SNE of the selected assays' results were illustrated.

**7-plex validation**. Following the 3-plex validation, seven targets were used to further validate the Smart-Plexer performance, where each target had at least two different assays, resulting in a total of 24 singleplexes and 4608 candidate combinations. Unlike for 3-plex, the mass number of combinations makes it impossible to empirically test all the assays in multiplex settings. Instead, representative groups of assays were chosen for the laboratory validation. Following the aforementioned Smart-Plexer workflow, after calculating simulated ADS and MDS on "$c$" parameters, six highest ranked (BEST) and six lowest ranked (BOT) combinations were picked out using the Ranking System. In addition, six middle-distant combinations (MID) were selected following the steps described in Table 4 (Algorithm 2).

TOP-ADS and TOP-MDS ($N = 6$) assays were selected empirically with large ADS but small MDS, and large MDS but small ADS, respectively. Similarly to the 3-plex validation, the relationship between simulated and empirical scores of the selected assays was explored by correlations of simulated and empirical metrics and comparisons of "$c$" parameter distributions. ACA was also applied to different groups of combinations. The complete pipeline of the 7-plex validation is illustrated in Supplementary Fig. 5.

**Clinical isolates classification**. To verify the feasibility of Smart-Plexer in real clinical settings, we chose the optimal multiplex assay (PM7.2151) that achieved the highest ACA accuracy in synthetic DNA testing and conducted experiments on clinical samples. The multiplex was tested on the clinical samples using qdPCR, with 770 unprocessed raw amplification data (including flat curves) as the output for each sample. After the data processing step, the curves were input into an ACA classifier pre-trained with synthetic DNA data, and curve-level predictions were assigned to every positive curve. The target category of a sample was then decided by finding the mostly shown label among all the sample's curve predictions. The confidence level of prediction is defined as the percentage of curves with this most shown label. Correctly predicted samples are marked as "detected", otherwise "undetected".

**Reporting summary**. Further information on research design is available in the Nature Portfolio Reporting Summary linked to this article.

## Data availability

The source data to generate Figs. 2–5 plots can be found in Supplementary Data. Additional data entire rank of combinations generated from singleplex assay are available

at https://github.com/LMigliet/SmartPlexer. Any remaining information can be obtained from the corresponding author upon reasonable request.

## Code availability

Code used to generate intermediate or final data and figures is available for download from the following archived code repository: https://github.com/LMigliet/SmartPlexer.

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

## Acknowledgements

This work was supported by the Imperial COVID-19 Research Fund (WDAI.G28059); the Department of Health and Social Care-funded Centre for Antimicrobial Optimisation (CAMO) at Imperial College London; the Imperial College President's PhD Scholarships 2021 (K.X.); a research scholarship by bioMérieux (L.K.); and the Imperial College's Centre for Antimicrobial Optimisation (CAMO). Authors F.B., K.H.C., A.H., P.G. and J.R.M. are affiliated with the NIHR Health Protection Research Unit (HPRU) in Healthcare Associated Infections and Antimicrobial Resistance at Imperial College London in partnership with the UK Health Security Agency (previously PHE) in collaboration with, Imperial Healthcare Partners, the University of Cambridge and the University of Warwick. A.H. is a National Institute for Health Research Senior Investigator.

## Author contributions

L.M. and J.R.M. conceived, designed and supervised the project. L.M. and Y.C., developed the algorithm and performed the data analysis. L.M., Z.L. and N.D. performed wet-lab experiments. K.X. review algorithm and data analytics concepts. T.P., A.M., L.K., M.C.S., A.H. and P.G. contributed to the proofreading and editing of the manuscript. L.M., K.X. and J.R.M. wrote the manuscript with input from all the authors.

## Competing interests

The authors declare no competing interests.
