## [Peer Review File · Communications Biology]

Reviewers' comments:

Reviewer #1 (Remarks to the Author):

In this study, the authors present an innovative framework, Smart-Plexer which is also a powerful approach for multiplex assay development. This new mathematical algorithm can simulate possible multiplex assay combinations based on singleplex real-time digital PCR data. Smart-Plexer can also overcome the challenges of constructing different amplification curve shapes for each multiplexed target.

I am happy to see the combination of wet-lab experiments and computational algorithms which is a complementary approach in the field. I congratulate the authors for the designation and utilization of this workflow. I predict this approach will open new horizons for novel methodologies in this area.

I have no concerns about the manuscript. The manuscript is well-organized and written. I recommend it for Communications Biology.

Reviewer #2 (Remarks to the Author):

The manuscript presents a novel approach to multiplex dPCR by employing a machine learning method (ACA) and developing an algorithm for identifying optimal primer set combinations. This represents a promising advance in the field. However, several aspects of the research require further clarification to enhance the reader's understanding.

In Results, the Smart-Plex design framework section, the distinction between the simulated multiplex and empirical multiplex needs elaboration. The section would benefit from a more comprehensive explanation of ACA. For the Smart-Plex framework, a multiplex test contains a primer set and all N_t targets, then authors need to train an ACA and classify amplification curves for these N_t targets in a single channel for further evaluation, like the transition of "c". However, the process by which the ground truth in the multiplex tests is determined remains unclear. The manuscript should address whether variations in target concentration affect the shape of the sigmoidal amplification curves. Are there discernible differences between amplification curves in different microwells?

In the Smart-Plexer validation using a 3-plex assay section, it would be helpful to have raw amplification curves complete with error bars included in the supplementary materials to demonstrate the assay's stability and repeatability.

In figure 2a, could you explain why the maximum value of the raw curve ADS is approximately 3.5, while the fitted parameters ADS can reach 70? In figure 2b, should the title of the left plot "Raw curve (ADS)" be "Raw curve (MDS)"?

In figure 3, the figure appears to lack color labels in the legend. To facilitate comparison, it is suggested to plot the distribution curves (simulated multiplex and empirical multiplex) for each target within the same figure.

In Results, the The key parameter for curve distances correlation in multiplex assays section, authors use RMSE to quantify difference between distributions of simulated and empirical multiplex. However, is RMSE a suitable criterion for evaluating difference between distributions? There are some criteria designed specifically for this task, like KL divergence.

In Methods, the data preprocessing section, how did authors decide the threshold value N_{zc} ? What is "AC count" in Table 1?

The manuscript should further elucidate the core advantages of this work compared to other multiplex dPCR methods with quantification capabilities, such as those detailed in DOI: 10.1039/D1AN01916C, 10.1039/D2LC00637E, and 10.1101/2023.05.10.540190.

Reviewer #3 (Remarks to the Author):

Developing multiple PCR assays requires a large number of experimental tests, the number of which grows exponentially with the number of multiple targets. Efforts must be made to establish appropriate multi-assay methods to ensure accurate and sensitive detection of large numbers of analytes in a single well response. Inspired by a data-driven approach, this paper reinvents the

process of generating and building multiway detections with Smart-Plexer, a hybrid, simple program that combines empirical testing of single-way detections with computer simulations to generate the best multiway combinations. Smart-Plexer developed an optimal set of multiplex PCR primers for reliable multi-pathogen identification by analyzing the dynamic target-to-target distance on the amplification curve. In addition, in this manuscript, the Smart-Plexer technique was effectively applied and assessed to identify seven respiratory illness targets using an improved multiplex PCR test.

Although the design method may still have some problems to be improved, the multi-amplification detection method invented in this manuscript has important guiding significance for the current clinical detection which needs high throughput, multiplexing and sensitive detection methods. The method presented in this paper provides a practical platform for cost-effective, rapid and accurate detection of pathogenic microbial infections and other nucleic acid multiple detection needs. The only shortcoming of this paper is that the practical application effect needs to be tested. The reviewers believe that the method established in this paper requires a large number of other multiple QPCR data for training and feedback improvement. Applications that are usually derived from just one data set will always have problems that need to be fixed in practice, although the author of this manuscript has conducted several validations. But anyway, the method established in this manuscript has important reference value for the successful establishment of nucleic acid multiple detection, the reviewer recommend that this manuscript can be accepted.

Imperial College London

5th July 2023

Dear Reviewers,

We are delighted you appreciate our methodology, and we would like to thank you for sharing very insightful comments. Please find below our responses to each of the point raised accompanying with our revised manuscript.

The main changes which address the reviewer's comments are highlighted in blue in the revised manuscript file.

We look forward to your response in due course.

Sincerely,

Jesus Rodriguez Manzano

Lecturer in Antimicrobial Resistance and Infectious Diseases

REVIEWER #1

Comment:

In this study, the authors present an innovative framework, Smart-Plexer which is also a powerful approach for multiplex assay development. This new mathematical algorithm can simulate possible multiplex assay combinations based on singleplex real-time digital PCR data. Smart-Plexer can also overcome the challenges of constructing different amplification curve shapes for each multiplexed target.

I am happy to see the combination of wet-lab experiments and computational algorithms which is a complementary approach in the field. I congratulate the authors for the designation and utilization of this workflow. I predict this approach will open new horizons for novel methodologies in this area.

I have no concerns about the manuscript. The manuscript is well-organized and written. I recommend it for Communications Biology.

Answer:

We express our gratitude to Reviewer 1 for the positive feedback and recommendation to publish our work in Communications Biology. We sincerely appreciate your valuable support.

We have made significant efforts to ensure clarity and coherence in presenting our research findings and methodology. We remain committed to further advancing the field through our research and hope that our work will contribute to the development of novel methodologies.

REVIEWER #2

Comment 1:

The manuscript presents a novel approach to multiplex dPCR by employing a machine learning method (ACA) and developing an algorithm for identifying optimal primer set combinations. This represents a promising advance in the field. However, several aspects of the research require further clarification to enhance the reader's understanding.

Answer 1:

We acknowledge the significance of addressing the concerns raised by Reviewer 2 as it plays a crucial role in enhancing the quality and clarity of our work. Please see relevant answers and edits below.

Comment 2.1:

In Results, the Smart-Plex design framework section, the distinction between the simulated multiplex and empirical multiplex needs elaboration. The section would benefit from a more comprehensive explanation of ACA.

Answer 2.1:

We have added the following text for clarification:

a) Page 2 line 73

“These simulations, representing different singleplex assay combinations, are further validated through wet-lab multiplex tests conducted for each target. These tests enable us to examine how the amplification reaction curve changes when transitioning from a singleplex to a multiplex environment. Therefore, empirical multiplex tests involve running the actual multiplex assay in the laboratory using the selected simulated multiplex primer sets. This allows us to directly observe and analyse the real-time amplification curves and evaluate the performance and accuracy of the multiplex assay (empirical multiplex), providing empirical validation of our approach when multiple primer sets are present.”

b) Page 2 line 82

“It is a methodology that utilizes machine learning techniques to analyse and classify the amplification curves generated in PCR reactions. By capturing the kinetic information encoded in the amplification curves, the ACA can effectively differentiate and identify different targets or analytes present in the reaction. It involves extracting relevant features from the amplification curves and employing machine learning algorithms to train a classifier that can accurately classify and distinguish between various targets based on their unique curve characteristics.”

Comment 2.2:

For the Smart-Plex framework, a multiplex test contains a primer set and all N_t targets, then authors need to train an ACA and classify amplification curves for these N_t targets in a single channel for further evaluation, like the transition of “c”. However, the process by which the ground truth in the multiplex tests is determined remains unclear.

Answer 2.2:

We thank the Reviewer for raising the question about the determination of the ground truth in our study.

In our research, the ground truth refers to the true labels or identities of the targets in the multiplex tests. For the synthetic DNA targets, we meticulously design and order specific sequences with known identities. These synthetic DNA targets serve as a reliable reference in the laboratory experiments. By incorporating them into the multiplex assays, we can precisely determine the expected targets and their corresponding amplification curves. In addition to synthetic DNA, we also utilize commercially available samples in our study, derived from real-world cases and contain known pathogens. The identification of the targets within these clinical samples is accomplished through manufacture reference. This reference information allows us to establish the ground truth for the evaluation of our approach.

To assess the performance of the ACA classifier, we compare its predictions with the established ground truth. By analysing the agreement between the predicted targets and the known identities, we can evaluate the accuracy and reliability of the ACA classifier in distinguishing different targets based on their unique amplification curve characteristics. To clarify this, we added the following.

Page 19 Line 750:

“To establish an accurate reference, we employed synthetic DNA targets with known identities and clinical samples with confirmed pathogen information. These reliable references were utilized to evaluate the ACA classifier and validate the effectiveness of the Smart-Plexer framework.

Comment 3:

The manuscript should address whether variations in target concentration affect the shape of the sigmoidal amplification curves. Are there discernible differences between amplification curves in different microwells?

In the Smart-Plexer validation using a 3-plex assay section, it would be helpful to have raw amplification curves complete with error bars included in the supplementary materials to demonstrate the assay's stability and repeatability.

Answer 3:

We acknowledge the reviewer's concern regarding the variations in the amplification curves and their potential impact on the sigmoidal shape. However, based on our study and the comprehensive data presented in the manuscript, we have observed that the slope of the PCR sigmoid (referred to as the "c" parameter) remains consistent, regardless of variations in the starting point of the exponential phase within different wells.

While amplification curves may exhibit variability (especially in Ct values), in our study we found that the "c" parameter remains a reliable indicator of the sigmoidal shape (please see Supplementary Fig. 2). Therefore, when comparing simulated multiplex assays to empirical ones, we specifically focus on the "c" parameter and do not consider other parameters, as it provides a robust measure for assessing the correlation and performance between the two.

Furthermore, we would like to emphasize that this preliminary work was conducted using digital PCR, and our primary objective was to identify different amplification curve shapes to increase the likelihood of optimal distances and differences between curves from different targets. We are

currently undertaking additional studies to validate the technology across platforms and expand our clinical sample cohort to further demonstrate its efficacy in clinical applications.

To address the reviewer's request, we have included an additional figure in the supplementary materials (**Supplementary Fig. 1** showing raw data) as well as an additional table (**Supplementary Table 3**) providing additional insights regarding the 3-plex experiments:

Page 4 Line 176:

Supplementary Fig. 1 illustrates the raw curve considered in this experiment and **Supplementary Table 3** shows the curve counts and the Ct variation among them.

Supplementary Fig. 1 | Raw curve from Smart-Plexer validation using a 3-plex assay.

Supplementary Table 3 | Raw curve details from Smart-Plexer validation using a 3-plex assay.

Target	Assay	Curve Counts	C _t values Means	C _t STD	Minimum C _t	Maximum C _t
Adenovirus	HAdV_HEX_03	639	24.66	1.43	19.12	40.57
Adenovirus	HAdV_HEX_09	563	27.50	1.10	23.78	31.16
Adenovirus	HAdV_HEX_12	581	26.48	1.15	23.00	30.37
Coronavirus HKU1	HCoV-HKU1_N_02	606	24.39	0.99	21.39	28.82
Coronavirus HKU1	HCoV-HKU1_N_04	598	24.47	1.06	21.72	28.45
Coronavirus HKU1	HCoV-HKU1_N_06	149	26.47	0.92	24.64	29.53
MERS	MERS-CoV_N_01	666	25.49	1.02	22.51	30.95
MERS	MERS-CoV_N_03	386	27.00	0.99	24.63	30.93
MERS	MERS-CoV_N_04	744	24.07	1.00	21.89	28.96

Comment 4.1:

In figure 2a, could you explain why the maximum value of the raw curve ADS is approximately 3.5, while the fitted parameters ADS can reach 70?

Answer 4.1:

We appreciate the reviewer's observation and thank them for bringing this up. The discrepancy in the maximum values between the raw curve ADS and the fitted parameters ADS in Figure 2a is due to the nature of the data.

In the case of the raw curves, all of them exhibit sigmoidal-like shapes, with the final fluorescence values being less than 5 in our experimental setup. Therefore, the maximum value for the raw curve ADS is approximately 3.5, reflecting the range of the fluorescence values observed. On the other hand, the fitted parameters can have larger values, such as ~100 for parameter "e". This discrepancy arises because the fitted parameters are not constrained by the same limits as the raw curve data. Consequently, certain parameters, like "e", may have higher values, leading to larger ADS values in the fitted parameter analysis.

For instance, in the 3-plex assay (PM3.12) involving targets ADE_HEX_09, CHK_N_02, and MER_N_04, the ADS of the raw curve is 2.67, while the ADS of the fitted parameters is 66.97. This significant difference can be attributed to the high values of parameter "e" for ADE_HEX_09, which are around 100, compared to the values below or near 20 for the other two assays. This discrepancy in the parameter values contributes to the larger ADS value in the fitted parameter analysis.

Comment 4.2:

In figure 2b, should the title of the left plot “Raw curve (ADS)” be “Raw curve (MDS)”?

Answer 4.2:

We thank the reviewer for spotting this mistake in the picture, which has now been corrected:

a) ADS of 3-plex experiment

b) MDS of 3-plex experiment

c) Curve type of 3-plex experiment

Comment 5.1:

In figure 3, the figure appears to lack color labels in the legend. To facilitate comparison, it is suggested to plot the distribution curves (simulated multiplex and empirical multiplex) for each target within the same figure.

Answer 5.1:

We appreciate the reviewer's suggestion to enhance the clarity of Figure 3. We agree that this would facilitate comparison between the distribution curves of simulated multiplex and empirical multiplex for each target.

To aid understanding and provide a comprehensive visualization with legends, we have included an additional figure in the supplementary data that depicts the distribution curves for each target within the same plot:

Page 8 Line 265:

Additionally, to facilitate visualisation of this distributions plot, **Supplementary Fig. 3** displays distribution of "c" of the simulated and empirical multiplex for each target.

Combination: **PM3.01** - RMSE: **0.022**

Combination: **PM3.07** - RMSE: **0.02**

Combination: **PM3.12** - RMSE: **0.019**

Supplementary Fig. 4. | Distribution curves of simulated multiplex and empirical multiplex for each target. The figure displays the distribution “c” of the simulated and empirical multiplex for each target. The x-axis represents the parameter values, while the y-axis indicates the frequency.

Comment 5.2:

In Results, the key parameter for curve distances correlation in multiplex assays section, authors use RMSE to quantify difference between distributions of simulated and empirical multiplex. However, is RMSE a suitable criterion for evaluating difference between distributions? There are some criteria designed specifically for this task, like KL divergence.

Answer 5.2:

We thank the reviewer for suggesting the use of KL divergence as an alternative criterion for evaluating the difference between distributions. While KL divergence is a widely used measure for comparing probability distributions, we have carefully considered its applicability in our specific project.

In our analysis, we have observed that the distributions of the “c” parameters do not exhibit significant noise and are not always strictly Gaussian in nature. Additionally, our primary objective is to assess the relative distance between the target values rather than determining the similarity of the actual distribution shapes.

Given these considerations, we have found that RMSE (Root Mean Square Error) provides a suitable and effective metric for our specific needs. RMSE measures the overall difference between two distributions, taking into account both the magnitude and direction of the differences between corresponding data points. It allows us to quantify the dissimilarity between simulated and empirical multiplexes in terms of the “c” parameter values and effectively capture the discrepancies between them.

While KL divergence is a valuable criterion in many scenarios, we have found that RMSE better aligns with the specific objectives and characteristics of our study. We have carefully validated and

interpreted our results using RMSE and believe it provides meaningful insights into the performance of our multiplex assay.

Thank you for highlighting this point, and we hope this explanation clarifies our rationale for using RMSE as the criterion for evaluating the difference between distributions in our analysis. Moreover, we add a sentence regarding this in the main manuscript:

Page 7 Line 248

The computed RMSE values for all the 3-plex combinations ranged from 0.003 to 0.050, which are negligible in comparison to the range of the "c" parameters. Given that the "c" parameter distributions in our study exhibit minimal noise and little variation in shape, and considering that our primary focus was on assessing relative distance rather than distribution similarity, RMSE proved to be a reliable criterion for evaluating the differences between distributions.

Comment 6.1:

In Methods, the data preprocessing section, how did authors decide the threshold value N_{zc} ?

Answer 6.1:

We appreciate the reviewer's comment and apologize for the oversight in not providing the reference for the threshold value N_{zc} in the data preprocessing section. The threshold value N_{zc} was determined based on the concept described in Miglietta et al. 2022 (Adaptive Filtering Framework to Remove Nonspecific and Low-Efficiency Reactions in Multiplex Digital PCR Based on Sigmoidal Trends - [doi/10.1021/acs.analchem.2c01883](https://doi.org/10.1021/acs.analchem.2c01883)). We have now included the reference in the manuscript (ref 39, Page 18 Line 658). Thank you for bringing this to our attention.

Comment 6.2:

What is "AC count" in Table 1?

Answer 6.2:

We thank the reviewer to spot this, as it can induce into confusion. We added the following at the end of the table: * AC count: Positive Amplification Curve count.

Comment 7:

The manuscript should further elucidate the core advantages of this work compared to other multiplex dPCR methods with quantification capabilities, such as those detailed in DOI: 10.1039/D1AN01916C, 10.1039/D2LC00637E, and 10.1101/2023.05.10.540190.

Answer 7:

We would like to express our gratitude to the reviewer for their insightful comment and for sharing the interesting publications.

The primary focus of our work is not on the quantification capabilities, as we utilize the Biomark HD system from Fluidigm for digital PCR. The system relies on partitioning for quantification and follows a Poisson distribution. The quantification in digital pattern can be achieved by counting the wells and correcting the counts using the Poisson distribution, as instructed by the manufacturer. When the

panels become saturated, alternative methods such as the one described in Miglietta et al 2021 can be utilised (<https://doi.org/10.3389/fmolb.2021.775299>). Additionally, our study encompasses a wide range of sample concentrations, and our main objective is to detect and recognize the target regardless of its concentration. We acknowledge the importance of quantification, especially in pathogen detection, and we utilize digital PCR instruments to ensure robust quantification.

Regarding the core advantages of our work compared to other multiplex dPCR methods discussed in the referenced articles:

1. DOI: 10.1101/2023.05.10.540190: This work introduces a novel approach called digital PCR by colour combination, enabling high-plex assays (up to 15-plex) by detecting each target with a combination of TaqMan probes and colours. However, it's important to note that this study utilizes a system that can only read FAM and uses ROX as a passive reference, allowing the authors to perform multiplexing of 7 targets in a single channel. This contrasts with our study, where we do not require instrumentation with a higher number of colour channels. The key benefit of our technology is two-fold: first, there is no need for expensive camera instrumentation with multiple fluorescent channels, and second, if multiple colours are available, the number of targets that can be multiplexed in a single reaction is multiplied by the number of colours. In our study, we achieve multiplexing of 7 targets in a single fluorescent channel, and if we consider 6 channels (as in the study mentioned), we potentially reach a multiplex of $7 \times 6 = 42$ targets.
2. DOI: 10.1039/d1an01916c: This article describes a multiplex digital PCR method using digital melting curve analysis (digital MCA) with a SlipChip microfluidic system. The digital MCA utilizes the unique melting temperature (T_m) of amplification products to classify positive partitions into different subgroups, enabling simultaneous digital quantification of multiple targets without limitations in fluorescence channels. However, it is important to note that this approach is suitable only for intercalating dye chemistries. In our previous work (<https://doi.org/10.1021/acs.analchem.0c03298>), we demonstrated a similar concept where we detected 9 targets in a single well and a single fluorescence channel reaction. However, we discovered limitations related to melting curve T_m distribution and the limited use of multiple fluorescent channels, which may hinder the true potential of the ACA technique. Our ACA method, on the other hand, allows for improved multiplexing capabilities without relying on melting curve analysis, making it advantageous for devices where the capability of generating melting curves is absent (e.g. Point-of-Care devices). Therefore, the main advantage of our method compared to the aforementioned work is (write something).
3. DOI: 10.1039/d2lc00637e: The referenced work introduces a deep learning-based similar colour analysis method (SCAD) for multiplex dPCR in a single fluorescent channel. This is an intriguing concept, especially as we are currently developing a high-level multiplexing approach using three different fluorescent channels. We encountered challenges in using closely-wavelength fluorescent channels as they tend to leak into the emission band, making it difficult to differentiate targets based on TaqMan fluorophore tags. However, the key difference is that they still utilize two different probes to tag and recognize the target. With our ACA method, if we use two fluorophores in TaqMan, we can detect $7 \times 2 = 14$ targets in a single well. This technique mentioned by the article has significant potential in complementing our technology to further increase the level of multiplexing. The technologies are different and offer benefits on different scales.

In summary, our work offers distinct advantages compared to the referenced articles. Our approach focuses on target recognition independent of concentration, and our ACA method improves

multiplexing capabilities without relying on melting curve analysis or multiple fluorescent channels. However, we state that for future applications all that aspect will be considered. Moreover, we believe that each of the abovementioned technology contributes to the field of multiplex dPCR and can be extremely beneficial for the evolution of the Smart-Plexer and the ACA method. Therefore, we added the suggested references in the conclusion section where we explain future application and limitations.

Page 14 Line 476:

By leveraging the optical capability of real-time PCR instruments, a multiplex assay using multiple-channel detection can double or triple the number of targets in a single reaction⁴⁵⁻⁴⁷. All these strategies aim to improve the ACA classification through a more innovative development from the chemistry perspective, while from the machine learning view, the current classifiers rely on state-of-the-art algorithms which shine for their robustness but are limited for tailoring to specific datasets.

REVIEWER #3

Comment:

Developing multiple PCR assays requires a large number of experimental tests, the number of which grows exponentially with the number of multiple targets. Efforts must be made to establish appropriate multi-assay methods to ensure accurate and sensitive detection of large numbers of analytes in a single well response. Inspired by a data-driven approach, this paper reinvents the process of generating and building multiway detections with Smart-Plexer, a hybrid, simple program that combines empirical testing of single-way detections with computer simulations to generate the best multiway combinations. Smart-Plexer developed an optimal set of multiplex PCR primers for reliable multi-pathogen identification by analyzing the dynamic target-to-target distance on the amplification curve. In addition, in this manuscript, the Smart-Plexer technique was effectively applied and assessed to identify seven respiratory illness targets using an improved multiplex PCR test.

Although the design method may still have some problems to be improved, the multi-amplification detection method invented in this manuscript has important guiding significance for the current clinical detection which needs high throughput, multiplexing and sensitive detection methods. The method presented in this paper provides a practical platform for cost-effective, rapid and accurate detection of pathogenic microbial infections and other nucleic acid multiple detection needs. The only shortcoming of this paper is that the practical application effect needs to be tested. The reviewers believe that the method established in this paper requires a large number of other multiple QPCR data for training and feedback improvement. Applications that are usually derived from just one data set will always have problems that need to be fixed in practice, although the author of this manuscript has conducted several validations. But anyway, the method established in this manuscript has important reference value for the successful establishment of nucleic acid multiple detection, the reviewer recommend that this manuscript can be accepted.

Answer:

We thank the Reviewer 3 for the positive assessment of our work and the recognition of the importance of high-throughput, multiplexing, and sensitive detection methods in clinical diagnostics.

We acknowledge your comment regarding the potential for further improvement in our design method. We agree that ongoing refinement and optimization of the Smart-Plexer technique are essential for its practical application in clinical settings. While we have conducted several validations to demonstrate the effectiveness of our approach, we understand the need for a larger dataset for training and feedback improvement. We are actively working on expanding the scope of our experimental validations to address this concern and ensure the reliability and robustness of the method.

We share your enthusiasm for the potential practical applications of our approach, particularly in cost-effective, rapid, and accurate detection of pathogenic microbial infections and other nucleic acid multiple detection needs. We sincerely appreciate your recommendation for acceptance of our manuscript. We will continue to advance our methodology based on your suggestions and insights.

To ensure that other readers gain insights and address concerns similar to those raised by Reviewer 3, we have added a sentence in the manuscript to provide further clarification. The sentence we have included is as follows:

Page 14 Line 490:

“To further validate the practical application of the Smart-Plexer method, future work using different platform (such as qPCR) and larger clinical samples cohort is currently in progress to ensure that the Smart-Plexer meets the requirements of high-throughput multiplexing, and sensitive detection methods in clinical diagnostics. By continuously refining and optimizing the methodology, we aim to establish a practical platform that offers cost-effective, rapid, and accurate detection of pathogenic microbial infections and other nucleic acid multiple detection needs.”

REVIEWERS' COMMENTS:

Reviewer #2 (Remarks to the Author):

There are no more comments.

Imperial College London

5th July 2023

Dear Reviewers,

We are delighted you appreciate our methodology, and we would like to thank you for sharing very insightful comments. Please find below our responses to each of the point raised accompanying with our revised manuscript.

The main changes which address the reviewer's comments are highlighted in blue in the revised manuscript file.

We look forward to your response in due course.

Sincerely,

Jesus Rodriguez Manzano
Lecturer in Antimicrobial Resistance and Infectious Diseases

REVIEWER #1

Comment:

In this study, the authors present an innovative framework, Smart-Plexer which is also a powerful approach for multiplex assay development. This new mathematical algorithm can simulate possible multiplex assay combinations based on singleplex real-time digital PCR data. Smart-Plexer can also overcome the challenges of constructing different amplification curve shapes for each multiplexed target.

I am happy to see the combination of wet-lab experiments and computational algorithms which is a complementary approach in the field. I congratulate the authors for the designation and utilization of this workflow. I predict this approach will open new horizons for novel methodologies in this area.

I have no concerns about the manuscript. The manuscript is well-organized and written. I recommend it for Communications Biology.

Answer:

We express our gratitude to Reviewer 1 for the positive feedback and recommendation to publish our work in Communications Biology. We sincerely appreciate your valuable support.

We have made significant efforts to ensure clarity and coherence in presenting our research findings and methodology. We remain committed to further advancing the field through our research and hope that our work will contribute to the development of novel methodologies.

REVIEWER #2

Comment 1:

The manuscript presents a novel approach to multiplex dPCR by employing a machine learning method (ACA) and developing an algorithm for identifying optimal primer set combinations. This represents a promising advance in the field. However, several aspects of the research require further clarification to enhance the reader's understanding.

Answer 1:

We acknowledge the significance of addressing the concerns raised by Reviewer 2 as it plays a crucial role in enhancing the quality and clarity of our work. Please see relevant answers and edits below.

Comment 2.1:

In Results, the Smart-Plex design framework section, the distinction between the simulated multiplex and empirical multiplex needs elaboration. The section would benefit from a more comprehensive explanation of ACA.

Answer 2.1:

We have added the following text for clarification:

a) Page 2 line 73

“These simulations, representing different singleplex assay combinations, are further validated through wet-lab multiplex tests conducted for each target. These tests enable us to examine how the amplification reaction curve changes when transitioning from a singleplex to a multiplex environment. Therefore, empirical multiplex tests involve running the actual multiplex assay in the laboratory using the selected simulated multiplex primer sets. This allows us to directly observe and analyse the real-time amplification curves and evaluate the performance and accuracy of the multiplex assay (empirical multiplex), providing empirical validation of our approach when multiple primer sets are present.”

b) Page 2 line 82

“It is a methodology that utilizes machine learning techniques to analyse and classify the amplification curves generated in PCR reactions. By capturing the kinetic information encoded in the amplification curves, the ACA can effectively differentiate and identify different targets or analytes present in the reaction. It involves extracting relevant features from the amplification curves and employing machine learning algorithms to train a classifier that can accurately classify and distinguish between various targets based on their unique curve characteristics.”

Comment 2.2:

For the Smart-Plex framework, a multiplex test contains a primer set and all N_t targets, then authors need to train an ACA and classify amplification curves for these N_t targets in a single channel for further evaluation, like the transition of “c”. However, the process by which the ground truth in the multiplex tests is determined remains unclear.

Answer 2.2:

We thank the Reviewer for raising the question about the determination of the ground truth in our study.

In our research, the ground truth refers to the true labels or identities of the targets in the multiplex tests. For the synthetic DNA targets, we meticulously design and order specific sequences with known identities. These synthetic DNA targets serve as a reliable reference in the laboratory experiments. By incorporating them into the multiplex assays, we can precisely determine the expected targets and their corresponding amplification curves. In addition to synthetic DNA, we also utilize commercially available samples in our study, derived from real-world cases and contain known pathogens. The identification of the targets within these clinical samples is accomplished through manufacture reference. This reference information allows us to establish the ground truth for the evaluation of our approach.

To assess the performance of the ACA classifier, we compare its predictions with the established ground truth. By analysing the agreement between the predicted targets and the known identities, we can evaluate the accuracy and reliability of the ACA classifier in distinguishing different targets based on their unique amplification curve characteristics. To clarify this, we added the following.

Page 19 Line 750:

“To establish an accurate reference, we employed synthetic DNA targets with known identities and clinical samples with confirmed pathogen information. These reliable references were utilized to evaluate the ACA classifier and validate the effectiveness of the Smart-Plexer framework.

Comment 3:

The manuscript should address whether variations in target concentration affect the shape of the sigmoidal amplification curves. Are there discernible differences between amplification curves in different microwells?

In the Smart-Plexer validation using a 3-plex assay section, it would be helpful to have raw amplification curves complete with error bars included in the supplementary materials to demonstrate the assay's stability and repeatability.

Answer 3:

We acknowledge the reviewer's concern regarding the variations in the amplification curves and their potential impact on the sigmoidal shape. However, based on our study and the comprehensive data presented in the manuscript, we have observed that the slope of the PCR sigmoid (referred to as the "c" parameter) remains consistent, regardless of variations in the starting point of the exponential phase within different wells.

While amplification curves may exhibit variability (especially in Ct values), in our study we found that the "c" parameter remains a reliable indicator of the sigmoidal shape (please see Supplementary Fig. 2). Therefore, when comparing simulated multiplex assays to empirical ones, we specifically focus on the "c" parameter and do not consider other parameters, as it provides a robust measure for assessing the correlation and performance between the two.

Furthermore, we would like to emphasize that this preliminary work was conducted using digital PCR, and our primary objective was to identify different amplification curve shapes to increase the likelihood of optimal distances and differences between curves from different targets. We are

currently undertaking additional studies to validate the technology across platforms and expand our clinical sample cohort to further demonstrate its efficacy in clinical applications.

To address the reviewer's request, we have included an additional figure in the supplementary materials (**Supplementary Fig. 1** showing raw data) as well as an additional table (**Supplementary Table 3**) providing additional insights regarding the 3-plex experiments:

Page 4 Line 176:

Supplementary Fig. 1 illustrates the raw curve considered in this experiment and **Supplementary Table 3** shows the curve counts and the Ct variation among them.

Supplementary Fig. 1 | Raw curve from Smart-Plexer validation using a 3-plex assay.

Supplementary Table 3 | Raw curve details from Smart-Plexer validation using a 3-plex assay.

Target	Assay	Curve Counts	C _t values Means	C _t STD	Minimum C _t	Maximum C _t
Adenovirus	HAdV_HEX_03	639	24.66	1.43	19.12	40.57
Adenovirus	HAdV_HEX_09	563	27.50	1.10	23.78	31.16
Adenovirus	HAdV_HEX_12	581	26.48	1.15	23.00	30.37
Coronavirus HKU1	HCoV-HKU1_N_02	606	24.39	0.99	21.39	28.82
Coronavirus HKU1	HCoV-HKU1_N_04	598	24.47	1.06	21.72	28.45
Coronavirus HKU1	HCoV-HKU1_N_06	149	26.47	0.92	24.64	29.53
MERS	MERS-CoV_N_01	666	25.49	1.02	22.51	30.95
MERS	MERS-CoV_N_03	386	27.00	0.99	24.63	30.93
MERS	MERS-CoV_N_04	744	24.07	1.00	21.89	28.96

Comment 4.1:

In figure 2a, could you explain why the maximum value of the raw curve ADS is approximately 3.5, while the fitted parameters ADS can reach 70?

Answer 4.1:

We appreciate the reviewer's observation and thank them for bringing this up. The discrepancy in the maximum values between the raw curve ADS and the fitted parameters ADS in Figure 2a is due to the nature of the data.

In the case of the raw curves, all of them exhibit sigmoidal-like shapes, with the final fluorescence values being less than 5 in our experimental setup. Therefore, the maximum value for the raw curve ADS is approximately 3.5, reflecting the range of the fluorescence values observed. On the other hand, the fitted parameters can have larger values, such as ~100 for parameter "e". This discrepancy arises because the fitted parameters are not constrained by the same limits as the raw curve data. Consequently, certain parameters, like "e", may have higher values, leading to larger ADS values in the fitted parameter analysis.

For instance, in the 3-plex assay (PM3.12) involving targets ADE_HEX_09, CHK_N_02, and MER_N_04, the ADS of the raw curve is 2.67, while the ADS of the fitted parameters is 66.97. This significant difference can be attributed to the high values of parameter "e" for ADE_HEX_09, which are around 100, compared to the values below or near 20 for the other two assays. This discrepancy in the parameter values contributes to the larger ADS value in the fitted parameter analysis.

Comment 4.2:

In figure 2b, should the title of the left plot “Raw curve (ADS)” be “Raw curve (MDS)”?

Answer 4.2:

We thank the reviewer for spotting this mistake in the picture, which has now been corrected:

a) ADS of 3-plex experiment

b) MDS of 3-plex experiment

c) Curve type of 3-plex experiment

Comment 5.1:

In figure 3, the figure appears to lack color labels in the legend. To facilitate comparison, it is suggested to plot the distribution curves (simulated multiplex and empirical multiplex) for each target within the same figure.

Answer 5.1:

We appreciate the reviewer's suggestion to enhance the clarity of Figure 3. We agree that this would facilitate comparison between the distribution curves of simulated multiplex and empirical multiplex for each target.

To aid understanding and provide a comprehensive visualization with legends, we have included an additional figure in the supplementary data that depicts the distribution curves for each target within the same plot:

Page 8 Line 265:

Additionally, to facilitate visualisation of this distributions plot, **Supplementary Fig. 3** displays distribution of "c" of the simulated and empirical multiplex for each target.

Combination: **PM3.01** - RMSE: **0.022**

Combination: **PM3.07** - RMSE: **0.02**

Combination: **PM3.12** - RMSE: **0.019**

Supplementary Fig. 4. | Distribution curves of simulated multiplex and empirical multiplex for each target. The figure displays the distribution “c” of the simulated and empirical multiplex for each target. The x-axis represents the parameter values, while the y-axis indicates the frequency.

Comment 5.2:

In Results, the key parameter for curve distances correlation in multiplex assays section, authors use RMSE to quantify difference between distributions of simulated and empirical multiplex. However, is RMSE a suitable criterion for evaluating difference between distributions? There are some criterions designed specifically for this task, like KL divergence.

Answer 5.2:

We thank the reviewer for suggesting the use of KL divergence as an alternative criterion for evaluating the difference between distributions. While KL divergence is a widely used measure for comparing probability distributions, we have carefully considered its applicability in our specific project.

In our analysis, we have observed that the distributions of the “c” parameters do not exhibit significant noise and are not always strictly Gaussian in nature. Additionally, our primary objective is to assess the relative distance between the target values rather than determining the similarity of the actual distribution shapes.

Given these considerations, we have found that RMSE (Root Mean Square Error) provides a suitable and effective metric for our specific needs. RMSE measures the overall difference between two distributions, taking into account both the magnitude and direction of the differences between corresponding data points. It allows us to quantify the dissimilarity between simulated and empirical multiplexes in terms of the “c” parameter values and effectively capture the discrepancies between them.

While KL divergence is a valuable criterion in many scenarios, we have found that RMSE better aligns with the specific objectives and characteristics of our study. We have carefully validated and

interpreted our results using RMSE and believe it provides meaningful insights into the performance of our multiplex assay.

Thank you for highlighting this point, and we hope this explanation clarifies our rationale for using RMSE as the criterion for evaluating the difference between distributions in our analysis. Moreover, we add a sentence regarding this in the main manuscript:

Page 7 Line 248

The computed RMSE values for all the 3-plex combinations ranged from 0.003 to 0.050, which are negligible in comparison to the range of the "c" parameters. Given that the "c" parameter distributions in our study exhibit minimal noise and little variation in shape, and considering that our primary focus was on assessing relative distance rather than distribution similarity, RMSE proved to be a reliable criterion for evaluating the differences between distributions.

Comment 6.1:

In Methods, the data preprocessing section, how did authors decide the threshold value N_{zc}?

Answer 6.1:

We appreciate the reviewer's comment and apologize for the oversight in not providing the reference for the threshold value N_{zc} in the data preprocessing section. The threshold value N_{zc} was determined based on the concept described in Miglietta et al. 2022 (Adaptive Filtering Framework to Remove Nonspecific and Low-Efficiency Reactions in Multiplex Digital PCR Based on Sigmoidal Trends - [doi/10.1021/acs.analchem.2c01883](https://doi.org/10.1021/acs.analchem.2c01883)). We have now included the reference in the manuscript (ref 39, Page 18 Line 658). Thank you for bringing this to our attention.

Comment 6.2:

What is "AC count" in Table 1?

Answer 6.2:

We thank the reviewer to spot this, as it can induce into confusion. We added the following at the end of the table: * AC count: Positive Amplification Curve count.

Comment 7:

The manuscript should further elucidate the core advantages of this work compared to other multiplex dPCR methods with quantification capabilities, such as those detailed in DOI: 10.1039/D1AN01916C, 10.1039/D2LC00637E, and 10.1101/2023.05.10.540190.

Answer 7:

We would like to express our gratitude to the reviewer for their insightful comment and for sharing the interesting publications.

The primary focus of our work is not on the quantification capabilities, as we utilize the Biomark HD system from Fluidigm for digital PCR. The system relies on partitioning for quantification and follows a Poisson distribution. The quantification in digital pattern can be achieved by counting the wells and correcting the counts using the Poisson distribution, as instructed by the manufacturer. When the

panels become saturated, alternative methods such as the one described in Miglietta et al 2021 can be utilised (<https://doi.org/10.3389/fmolb.2021.775299>). Additionally, our study encompasses a wide range of sample concentrations, and our main objective is to detect and recognize the target regardless of its concentration. We acknowledge the importance of quantification, especially in pathogen detection, and we utilize digital PCR instruments to ensure robust quantification.

Regarding the core advantages of our work compared to other multiplex dPCR methods discussed in the referenced articles:

1. DOI: 10.1101/2023.05.10.540190: This work introduces a novel approach called digital PCR by colour combination, enabling high-plex assays (up to 15-plex) by detecting each target with a combination of TaqMan probes and colours. However, it's important to note that this study utilizes a system that can only read FAM and uses ROX as a passive reference, allowing the authors to perform multiplexing of 7 targets in a single channel. This contrasts with our study, where we do not require instrumentation with a higher number of colour channels. The key benefit of our technology is two-fold: first, there is no need for expensive camera instrumentation with multiple fluorescent channels, and second, if multiple colours are available, the number of targets that can be multiplexed in a single reaction is multiplied by the number of colours. In our study, we achieve multiplexing of 7 targets in a single fluorescent channel, and if we consider 6 channels (as in the study mentioned), we potentially reach a multiplex of $7 \times 6 = 42$ targets.
2. DOI: 10.1039/d1an01916c: This article describes a multiplex digital PCR method using digital melting curve analysis (digital MCA) with a SlipChip microfluidic system. The digital MCA utilizes the unique melting temperature (T_m) of amplification products to classify positive partitions into different subgroups, enabling simultaneous digital quantification of multiple targets without limitations in fluorescence channels. However, it is important to note that this approach is suitable only for intercalating dye chemistries. In our previous work (<https://doi.org/10.1021/acs.analchem.0c03298>), we demonstrated a similar concept where we detected 9 targets in a single well and a single fluorescence channel reaction. However, we discovered limitations related to melting curve T_m distribution and the limited use of multiple fluorescent channels, which may hinder the true potential of the ACA technique. Our ACA method, on the other hand, allows for improved multiplexing capabilities without relying on melting curve analysis, making it advantageous for devices where the capability of generating melting curves is absent (e.g. Point-of-Care devices). Therefore, the main advantage of our method compared to the aforementioned work is (write something).
3. DOI: 10.1039/d2lc00637e: The referenced work introduces a deep learning-based similar colour analysis method (SCAD) for multiplex dPCR in a single fluorescent channel. This is an intriguing concept, especially as we are currently developing a high-level multiplexing approach using three different fluorescent channels. We encountered challenges in using closely-wavelength fluorescent channels as they tend to leak into the emission band, making it difficult to differentiate targets based on TaqMan fluorophore tags. However, the key difference is that they still utilize two different probes to tag and recognize the target. With our ACA method, if we use two fluorophores in TaqMan, we can detect $7 \times 2 = 14$ targets in a single well. This technique mentioned by the article has significant potential in complementing our technology to further increase the level of multiplexing. The technologies are different and offer benefits on different scales.

In summary, our work offers distinct advantages compared to the referenced articles. Our approach focuses on target recognition independent of concentration, and our ACA method improves

multiplexing capabilities without relying on melting curve analysis or multiple fluorescent channels. However, we state that for future applications all that aspect will be considered. Moreover, we believe that each of the abovementioned technology contributes to the field of multiplex dPCR and can be extremely beneficial for the evolution of the Smart-Plexer and the ACA method. Therefore, we added the suggested references in the conclusion section where we explain future application and limitations.

Page 14 Line 476:

By leveraging the optical capability of real-time PCR instruments, a multiplex assay using multiple-channel detection can double or triple the number of targets in a single reaction⁴⁵⁻⁴⁷. All these strategies aim to improve the ACA classification through a more innovative development from the chemistry perspective, while from the machine learning view, the current classifiers rely on state-of-the-art algorithms which shine for their robustness but are limited for tailoring to specific datasets.

REVIEWER #3

Comment:

Developing multiple PCR assays requires a large number of experimental tests, the number of which grows exponentially with the number of multiple targets. Efforts must be made to establish appropriate multi-assay methods to ensure accurate and sensitive detection of large numbers of analytes in a single well response. Inspired by a data-driven approach, this paper reinvents the process of generating and building multiway detections with Smart-Plexer, a hybrid, simple program that combines empirical testing of single-way detections with computer simulations to generate the best multiway combinations. Smart-Plexer developed an optimal set of multiplex PCR primers for reliable multi-pathogen identification by analyzing the dynamic target-to-target distance on the amplification curve. In addition, in this manuscript, the Smart-Plexer technique was effectively applied and assessed to identify seven respiratory illness targets using an improved multiplex PCR test.

Although the design method may still have some problems to be improved, the multi-amplification detection method invented in this manuscript has important guiding significance for the current clinical detection which needs high throughput, multiplexing and sensitive detection methods. The method presented in this paper provides a practical platform for cost-effective, rapid and accurate detection of pathogenic microbial infections and other nucleic acid multiple detection needs. The only shortcoming of this paper is that the practical application effect needs to be tested. The reviewers believe that the method established in this paper requires a large number of other multiple QPCR data for training and feedback improvement. Applications that are usually derived from just one data set will always have problems that need to be fixed in practice, although the author of this manuscript has conducted several validations. But anyway, the method established in this manuscript has important reference value for the successful establishment of nucleic acid multiple detection, the reviewer recommend that this manuscript can be accepted.

Answer:

We thank the Reviewer 3 for the positive assessment of our work and the recognition of the importance of high-throughput, multiplexing, and sensitive detection methods in clinical diagnostics.

We acknowledge your comment regarding the potential for further improvement in our design method. We agree that ongoing refinement and optimization of the Smart-Plexer technique are essential for its practical application in clinical settings. While we have conducted several validations to demonstrate the effectiveness of our approach, we understand the need for a larger dataset for training and feedback improvement. We are actively working on expanding the scope of our experimental validations to address this concern and ensure the reliability and robustness of the method.

We share your enthusiasm for the potential practical applications of our approach, particularly in cost-effective, rapid, and accurate detection of pathogenic microbial infections and other nucleic acid multiple detection needs. We sincerely appreciate your recommendation for acceptance of our manuscript. We will continue to advance our methodology based on your suggestions and insights.

To ensure that other readers gain insights and address concerns similar to those raised by Reviewer 3, we have added a sentence in the manuscript to provide further clarification. The sentence we have included is as follows:

Page 14 Line 490:

“To further validate the practical application of the Smart-Plexer method, future work using different platform (such as qPCR) and larger clinical samples cohort is currently in progress to ensure that the Smart-Plexer meets the requirements of high-throughput multiplexing, and sensitive detection methods in clinical diagnostics. By continuously refining and optimizing the methodology, we aim to establish a practical platform that offers cost-effective, rapid, and accurate detection of pathogenic microbial infections and other nucleic acid multiple detection needs.”